# The Out-of-Distribution Problem in Explainability and Search Methods for Feature Importance Explanations

**Peter Hase, Harry Xie,** and **Mohit Bansal**
Department of Computer Science
University of North Carolina at Chapel Hill
{peter, fengyu.xie, mbansal}@cs.unc.edu

## Abstract

Feature importance (FI) estimates are a popular form of explanation, and they are commonly created and evaluated by computing the change in model confidence caused by removing certain input features at test time. For example, in the standard *Sufficiency* metric, only the top-$k$ most important tokens are kept. In this paper, we study several under-explored dimensions of FI explanations, providing conceptual and empirical improvements for this form of explanation. First, we advance a new argument for why it can be problematic to remove features from an input when creating or evaluating explanations: the fact that these counterfactual inputs are out-of-distribution (OOD) to models implies that the resulting explanations are *socially misaligned*. The crux of the problem is that the model prior and random weight initialization influence the explanations (and explanation metrics) in unintended ways. To resolve this issue, we propose a simple alteration to the model training process, which results in more socially aligned explanations and metrics. Second, we compare among five approaches for removing features from model inputs. We find that some methods produce more OOD counterfactuals than others, and we make recommendations for selecting a feature-replacement function. Finally, we introduce four search-based methods for identifying FI explanations and compare them to strong baselines, including LIME, Anchors, and Integrated Gradients. Through experiments with six diverse text classification datasets, we find that the only method that consistently outperforms random search is a Parallel Local Search (PLS) that we introduce. Improvements over the second best method are as large as 5.4 points for Sufficiency and 17 points for Comprehensiveness.[1]

## 1 Introduction

Estimating feature importance (FI) is a common approach to explaining how learned models make predictions for individual data points [51, 46, 34, 57, 36, 11]. FI methods assign a scalar to each feature of an input representing its "importance" to the model's output, where a feature may be an individual component of an input (such as a pixel or a word) or some combination of components. Alongside these methods, many approaches have been proposed for evaluating FI estimates (also known as attributions) [41, 1, 13, 22, 20, 67]. Many of these approaches use test-time input ablations, where features marked as important are removed from the input, with the expectation that the model's confidence in its original prediction will decline if the selected features were truly important.

---

[1]All supporting code for experiments in this paper is publicly available at https://github.com/peterbhase/ExplanationSearch.

35th Conference on Neural Information Processing Systems (NeurIPS 2021).

For instance, according to the *Sufficiency* metric [13], the best FI explanation is the set of features which, if kept, would result in the highest model confidence in its original prediction. Typically the top-$k$ features would be selected according to their FI estimates, for some specified *sparsity* level $k$. Hence, the final explanation $e$ is a $k$-sparse binary vector in $\{0,1\}^d$, where $d$ is the dimensionality of the chosen feature space. For an explanation $e$ and a model $f$ that outputs a distribution over classes $p(y|x) = f(x)$, Sufficiency can be given as:

$$\text{Suff}(f, x, e) = f(x)_{\hat{y}} - f(\texttt{Replace}(x, e))_{\hat{y}}$$

where $\hat{y} = \arg\max_y f(x)_y$ is the predicted class for $x$ and the $\texttt{Replace}$ function replaces features in $x$ with some uninformative feature at locations corresponding to 0s in the explanation $e$.

The $\texttt{Replace}$ function plays a key role in the definition of such metrics because it defines the *counterfactual* input that we are comparing the original input with. Though FI explanations are often presented without mention of counterfactuals, all explanations make use of counterfactual situations [38], and FI explanations are no exception. The only way we can understand what makes some features "important" to a particular model prediction is by reference to a counterfactual input which has its important features replaced with a user-specified (uninformative) feature.

In this paper, we study several under-explored dimensions of the problem of finding good explanations according to test-time ablation metrics including Sufficiency and a related metric, Comprehensiveness, with a focus on natural language processing tasks. We describe three primary contributions below.

First, we argue that standard FI explanations are heavily influenced by the out-of-distribution (OOD) nature of counterfactual model inputs, which results in *socially misaligned* explanations. We use this term, first introduced by Jacovi and Goldberg [24], to describe a situation where an explanation communicates a different kind of information than the kind that people expect it to communicate. Here, we do not expect the model prior or random weight initialization to influence FI estimates. This is a problem insofar as FI explanations are not telling us what we think they are telling us. We propose a training algorithm to resolve the social misalignment, which is to expose models to counterfactual inputs during training, so that counterfactuals are not out-of-distribution at test time.

Second, we systematically compare $\texttt{Replace}$ functions, since this function plays an important role in evaluating explanations. To do so, we remove tokens from inputs using several $\texttt{Replace}$ functions, then measure how OOD these ablated inputs are to the model. We compare methods that remove tokens entirely from sequences of text [41, 13], replace token embeddings with the zero embedding or a special token [34, 57, 1, 65, 57], marginalize predictions over possible counterfactuals [68, 30, 63], and edit the input attention mask rather than the input text. Following our argument regarding the OOD problem (Sec. 4), we recommend the use of some $\texttt{Replace}$ functions over others.

Third, we provide several novel search-based methods for identifying FI explanations. While finding the optimal solution to $\arg\max_e \text{Suff}(f, x, e)$ is a natural example of binary optimization, a problem for which search algorithms are a common solution [43, 49, 3], we are aware of only a few prior works that search for good explanations [19, 47, 15]. We introduce our novel search algorithms for finding good explanations by making use of general search principles [43]. Based on experiments with two Transformer models and six text classification datasets (including FEVER, SNLI, and others), we summarize our core findings as follows:

1. We propose to train models on explanation counterfactuals and find that this leads to greater model robustness against counterfactuals and yields drastic differences in explanation metrics.

2. We find that some $\texttt{Replace}$ functions are better than others at reducing counterfactual OOD-ness, although ultimately our solution to the OOD problem is much more effective.

3. We introduce four novel search-based methods for identifying explanations. Out of all the methods we consider (including popular existing methods), the only one that consistently outperforms random search is the Parallel Local Search (PLS) that we introduce, often by large margins of up to 20.8 points. Importantly, we control for the compute budget used by each method.

## 2 Related Work

**Feature Importance Methods.** A great number of methods have been introduced for FI estimation, drawing upon local approximation models [46, 47, 31], attention weights [25, 61, 66], model gradients [51, 50, 57, 52], and model-based feature selection [4, 2, 62, 42, 8, 11]. While search approaches are

regularly used to solve combinatorial optimization problems in machine learning [49, 5, 3, 16, 40], we know of only a few FI methods based on search [19, 47, 15]. Fong and Vedaldi [19] perform gradient descent in explanation space, while Ribeiro et al. [47] search for probably-sufficient subsets of the input (under a perturbation distribution). In concurrent work, Du and Xu [15] propose a genetic search algorithm for identifying FI explanations. We introduce several novel search algorithms for finding good explanations, including (1) a gradient search similar to Fong and Vedaldi [19], a (2) local heuristic search inspired by an adversarial attack method [16], (3) a global heuristic search, and (4) a parallel local search (PLS) making use of general search principles [43].

**Choice of `Replace` Function.** Past evaluations of explanation methods typically remove tokens or words from the text entirely [41, 13] or replace them with fixed feature values [22, 64]. Methods for creating explanations also use several distinct approaches, including (1) replacing token embeddings with the zero vector [34, 57, 1], (2) using a special token [65, 57], (3) marginalizing predictions over a random variable representing an unobserved token [68, 30, 63, 27], and (4) adversarially selecting counterfactual features [23]. Sturmfels et al. [55] carry out a case study involving a few `Replace` functions for a vision model, which they compare via test-time ablations with image blurring techniques, though the case study is not intended to be a full comparison of methods. Haug et al. [21] assess a number of `Replace` functions for explanation methods used with tabular data, but they compare between functions to use when generating explanations, rather than when evaluating explanations, for which they offer no recommendation. In addition to evaluating `Replace` functions from the above works, we also consider setting attention mask values for individual tokens to 0.

**The Out-of-distribution Problem of Explanations.** Many papers have expressed concerns over how removing features from an input may result in counterfactuals that are out-of-distribution to a trained model [65, 57, 19, 7, 22, 30, 23, 63, 56, 27, 45, 21, 48, 28, 59]. In response to the problem, some have proposed to marginalize model predictions over possible counterfactual inputs [68, 30, 63, 27], use counterfactuals close to real inputs [7, 48], weight their importance by their closeness to real inputs [45], or to adversarially select counterfactual features rather than use any user-specified features [23]. Others reject the whole notion of test-time ablations, preferring metrics based on train-time ablations [22]. Jethani et al. [28] propose a specialized model for evaluating explanations that is trained on counterfactual inputs in order to make them in-distribution, but since the evaluation model is distinct from the model used to solve the task, explanation metrics calculated using this evaluation model may not reflect the faithfulness of explanations to the task model. In concurrent work, Vafa et al. [59] independently adopt a solution equivalent to our Counterfactual Training with an Attention Mask `Replace` function, an approach which we empirically justify in Sec. 5. In general, prior works make arguments for their approach based on intuition or basic machine learning principles, such as avoiding distribution shift. In Sec. 4, we give a more detailed argument for preferring in-distribution counterfactuals on the basis of *social alignment*, a concept introduced by Jacovi and Goldberg [24], and we propose a solution to the OOD problem. Our solution allows for test-time evaluation of explanations of a particular model's decisions for individual data points, unlike similar proposals which either evaluate large sets of explanations all at once [22] or use a separate model trained specifically for evaluation rather than the blackbox model [28].

## 3 Problem Statement

**Feature Importance Metrics.** The problem we are investigating is to find good feature importance explanations for single data points, where explanation are evaluated under metrics using test-time ablations of the input. In this context, an explanation for an input in a $d$ dimensional feature space is a binary vector $e \in \{0, 1\}^d$, which may be derived from discretizing an FI estimate $v \in \mathbb{R}^d$. We consider two primary metrics, *Sufficiency* and *Comprehensiveness* [13]. Sufficiency measures whether explanations identify a subset of features which, when kept, lead the model to remain confident in its original prediction for a data point. Comprehensiveness, meanwhile, measures whether an explanation identifies all of the features that contribute to a model's confidence in its prediction, such that removing these features from the input lowers the model's confidence.

The Sufficiency metric for an explanation $e \in \{0, 1\}^d$ and model $p(y|x) = f_\theta(x)$ is given as

$$\text{Suff}(f_\theta, x, e, s) = f_\theta(x)_{\hat{y}} - f_\theta(\texttt{Replace}_s(x, e))_{\hat{y}} \tag{1}$$

where $\hat{y} = \arg\max_y f(x)_y$ is the predicted class for $x$, and the $\texttt{Replace}_s$ function retains a proportion $s$ of the input features (indicated by $e$) while replacing the remaining features with some

user-specified feature. In order to control for the explanation sparsity, i.e. the proportion $s$ of tokens in the input that may be retained, we average Sufficiency scores across sparsity levels in $\{.05, .10, .20, .50\}$, meaning between 5% and 50% of tokens in the input are retained [13]. A **lower** Sufficiency value is better, as it indicates that more of the model's confidence is explained by just the retained features (increasing $f_\theta(\texttt{Replace}(x, e))_{\hat{y}}$).

Similarly, Comprehensiveness is given as $\text{Comp}(f_\theta, x, e, s) = f_\theta(x)_{\hat{y}} - f_\theta(\texttt{Replace}_s(x, e))_{\hat{y}}$ but with sparsity values in $\{.95, .90, .80, .50\}$, as we are looking to remove features that are important to the prediction (while keeping most features). A **higher** Comprehensiveness value is better, as it indicates that the explanation selects more of the evidence that contributes to the model's confidence in its prediction (resulting in a lower $f_\theta(\texttt{Replace}(x, e))_{\hat{y}}$).

**Overall Objective.** Finally, our overall Sufficiency and Comprehensiveness objectives are given by averaging Suff (or Comp) scores across several sparsity levels. With a model $p(y|x) = f(x)$, a single data point $x$ with $d$ features, and a set of sparsity levels $S$, the Sufficiency objective is optimized by obtaining a set $E = \{e_i\}_{i=1}^{|S|}$ with one explanation per sparsity level as

$$\arg\max_{E} \frac{1}{|S|} \sum_{i=1}^{|S|} \text{Suff}(f, x, e_i, s_i) \quad \text{s.t. } e_i \in \{0, 1\}^d \text{ and } \sum_{d} e_i^{(d)} \leq \text{ceiling}(s_i \cdot d)$$

where the ceiling function rounds up the number $s_i \cdot d$ of tokens to keep. When optimizing for Comprehensiveness, we use the Comp and $\arg\min$ functions, and the inequality is flipped. In general, we will optimize this objective using a limited compute budget, further described in Sec. 6.2.

# 4 The Out-of-Distribution Problem in Explanations

In this section, we first give a full argument for why it is problematic for explanations to be created *or* evaluated using OOD counterfactuals. Then, we propose a solution to the OOD problem. We rely on this argument in our comparison of `Replace` functions in Sec. 5. We also assess our proposed solution to the OOD problem in Sec. 5 and later make use of this solution in Sec. 6.

The OOD problem for explanations occurs when a counterfactual input used to create or evaluate an explanation is out-of-distribution (OOD) to a model. Here, we take OOD to mean the input is not drawn from the data distribution used in training (or for finetuning, when a model is finetuned) [39]. In general, counterfactual data points used to produce FI explanations will be OOD because they contain feature values not seen in training, like a MASK token for a language model. A long line of past work raises concerns with this fact [65, 57, 19, 7, 22, 30, 23, 63, 56, 27, 45, 21, 48, 28, 59]. The most concrete argument on the topic originates from Hooker et al. [22], who observe that using OOD counterfactuals makes it difficult to determine whether model performance degradation is caused by "distribution shift" or by the removal of information. It is true that, for a given counterfactual, a model might have produced a different prediction if that counterfactual was in-distribution rather than out-of-distribution. But this is a question we cannot ask about a single, trained model, where there is no ambiguity about what causes a drop in model confidence when replacing features: the features in the input were replaced, and this changes *that model's* prediction. If the counterfactual was in-distribution, we would be talking about a different model, with a different training distribution. Hence, we believe we need a stronger argument for why we should not use OOD counterfactuals when explaining a trained model's behavior.

Our principal claim is that feature importance explanations for a standardly trained neural model are *socially misaligned*, which is undesirable. Jacovi and Goldberg [24] originally introduce this term as they describe shortcomings of explanations from a pipeline (select-predict) model, which is a kind of classifier designed to be interpretable. Explanations are socially misaligned when people expect them to communicate one kind of information, and instead they communicate a different kind of information. For example, if we expected an explanation to be the information that a model relied on in order to reach a decision, but the explanation was actually information selected after a decision was already made, then we would say that the explanations are socially misaligned. Our argument now proceeds in two steps: first, we outline what the social expectations are for feature importance explanations, and then we argue that the social expectations are violated due to the fact that counterfactuals are OOD.

We suggest that, for a particular trained model and a particular data point, **people expect a feature importance explanation to reflect how the model *has learned* to interpret features as evidence for or against a particular decision.**[2] This social expectation is upheld if FI explanations are influenced only by the interaction of an untrained model with a training algorithm and data. But our expectations are violated to the ex-

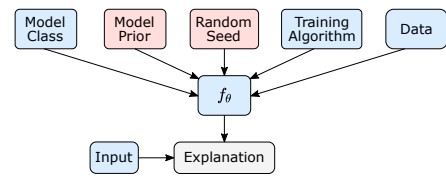

Figure 1: Causal diagram of a feature importance explanation for a trained model and an input.

tent that FI explanations are influenced by factors such as the choice of model prior and random seed (which we do not intend to influence FI explanations). We depict these possible upstream causes of individual FI explanations in Fig. 1. In fact, the model prior and random seed are influential to FI explanations when the counterfactuals employed in these explanations are OOD to the model. A simple example clearly illustrates the potential influence of model priors: Suppose one trained a BERT model to classify the sentiment of individual words using training data from a sentiment dictionary, then obtained feature importance explanations with the MASK token `Replace` function. In this situation, model predictions on counterfactual data are always equal to the prediction for a single MASK token, $f_\theta(\text{MASK})$. So, by construction, the MASK token never appears in the training data, but FI explanations for this model make use of the quantity $f_\theta(\text{MASK})$. Since a model could not have learned its prediction $f_\theta(\text{MASK})$ from the data, this quantity will be largely determined by the model prior and other training hyperparameters, and therefore explanations based on this prediction are socially misaligned. Now, in general, we know that neural models are sensitive to random parameter initialization, data ordering (determined by the random seed) [14], and hyperparameters (including regularization coefficients) [9, 44, 60], even as evaluated on in-distribution data. For OOD data, then, a neural model will still be influenced by these factors, but the model has no data to learn from in this domain. As a result, FI explanations are socially misaligned to the extent that these unexpected factors influence the explanations (while the expected factors like data are not as influential). In other words, we do not expect explanations to influenced by random factors, the priors of the model designer, or uninterpretable hyperparameters, but we do expect them to be influenced by what the model learns from data.

The argument applies equally to explanation metrics. When metrics are computed using OOD counterfactuals, the scores are influenced by unexpected factors like the model prior and random seed, rather than the removal of features that a model has learned to rely on. As a result, the metrics are socially misaligned. They do not represent explanation quality in the way we expect them to.

The solution to the OOD problem is to **align the train and test distributions, which we do by exposing the model to counterfactual inputs during training**, a method we term Counterfactual Training. Since common explanation methods can require hundreds or thousands of model forward passes when explaining predictions [46, 57], explanations from these methods would be prohibitively expensive to obtain during training. We therefore propose to train with random explanations that remove most of the input tokens, which provides a good objective in theory for models to learn the counterfactual distribution that will be seen at test time [28]. Specifically, we double the inputs in each training batch, adding a `Replace`$(x, e)$ version of each input (with the same label) according to a random explanation $e$ with sparsity randomly selected from $\{.05, .1, .2, .5\}$. The resulting Counterfactual-Trained (CT) models make in-distribution predictions for both observed and counterfactual data points. While we cannot guarantee that this approach fully eliminates the influence of the model prior and random seed on FI explanations, the fact that explanations are influenced by what the model learns from data will resolve social misalignment to a great extent. We find that these models suffer only slight drops in test set accuracy, by 0.7 points on average across six datasets (see Table **??** in Supplement). But we observe that this approach greatly improves model robustness to counterfactual inputs, meaning the counterfactuals are now much more in-distribution to models (described further in Sec. 5). Similar to the goals of ROAR [22] and EVAL-X [28], our proposed solution also aims to align the train and test-time distributions. However, our approach allows for test-time evaluation of individual explanations for a particular trained model, while ROAR only processes large sets of explanations all at once and EVAL-X introduces a specialized model for evaluation, which may not reflect the faithfulness of explanations to the task model.

---

[2]We mean "people" to refer to the typical person who has heard the standard description of these explanations, i.e. that they identify features that are "important" to a model decision. Of course, there will be some diversity in how different populations interpret feature importance explanations [18].

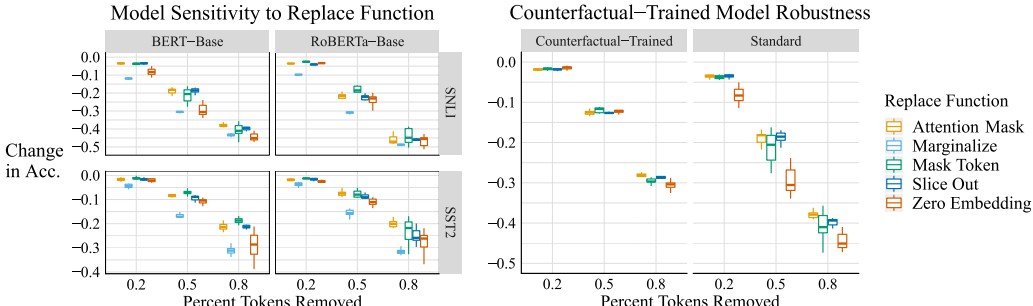

Figure 2: Model sensitivity to input ablations for several choices of `Replace` function and training algorithm. On the left we show the sensitivity of standardly trained models. On the right we show the effect of using Counterfactual-Trained models.

## 5 Analysis of Counterfactual Input OOD-ness

Here, we assess how out-of-distribution the counterfactual inputs given by `Replace` functions are to models, and we measure the effectiveness of Counterfactual Training. We do this before designing or evaluating explanation methods because, given our argument in Sec. 4, it is important to first identify which `Replace` function and training methods are most appropriate to use for these purposes.

**Experiment Design.** We compare between `Replace` functions according to how robust models are to test-time input ablations using each function, where the set of input features to be removed is fixed across the functions. We measure robustness by model accuracy, which serves as a proxy for how in-distribution or out-of-distribution the ablated inputs are. If we observe differences in model accuracies between `Replace` functions for a given level of feature sparsity, we can attribute the change in the input OOD-ness to the `Replace` function itself. In the same manner, we compare between Counterfactual-Trained (CT) models and standardly trained models (termed as Standard).

Specifically, we train 10 BERT-Base [12] and RoBERTa-Base [35] on two benchmark text classification datasets, SNLI [6] and SST-2 [54]. These are all Standard models, without the counterfactual training we propose. We use ten random seeds for training these models. Then, we evaluate how robust the models are to input ablations, where we remove a proportion of random tokens using one of five `Replace` functions (i.e. we `Replace` according to a random explanation). We evaluate across proportions in {0.2, 0.5, 0.8}. The five `Replace` functions we test are:

1. **Attention Mask.** We introduce this `Replace` function, which sets a Transformer's attention mask values for removed tokens to 0, meaning their hidden representations are never attended to.
2. **Marginalize.** This method marginalizes model predictions over counterfactuals drawn from a generative model $p_\phi(x)$ of the data, i.e. as $\arg\max_y \ln \sum_{\tilde{x} \sim p_\phi(\tilde{x}|x,e)} p_\theta(y|\tilde{x}) p_\phi(\tilde{x}|x,e)$, where $p_\phi(\tilde{x}|x,e)$ is the distribution over tokens to be replaced given by e.g. BERT [68, 30, 7, 63].
3. **MASK Token**. In this method, we simply replace tokens with the MASK token.
4. **Slice Out**. This approach removes selected tokens from the input sequence itself, such that the resulting sequence has a lower number of tokens.
5. **Zero Vector**. Here, we set the token embeddings for removed tokens to the zero vector.

We train additional CT models for BERT-Base on SNLI, with ten random seeds per model, for all `Replace` functions except Marginalize, since this function is exceedingly expensive to use during Counterfactual Training. For further details, see the Supplement.

**Results for `Replace` functions.** We show the results of this experiment in Fig. 2, via boxplots of the drops in accuracy for each of the 10 models per condition. First, we describe differences in `Replace` functions for Standard models, then we discuss the effect of Counterfactual Training. On the left in Fig. 2, we see that Standard models are much more sensitive to some `Replace` functions than others. The Attention Mask and Mask Token functions are the two best methods. The best of these two methods outperforms the third best method by up to 1.61 points with BERT and SNLI ($p = .0005$),[3]

---

[3] $p$-values for two-sided difference in means tests are calculated by block bootstrap with all data points and model seeds being resampled 100k times [17].

5.48 points with RoBERTa and SNLI ($p < 1\mathrm{e}{-}4$), 2.42 points with BERT and SST-2 ($p = 0.0008$), and 4.72 points with RoBERTa and SST-2 ($p < 1\mathrm{e}{-}4$). The other methods often far underperform the best method. For instance, with BERT on SST-2, Zero Embedding is up to 10.45 points worse than Mask Token ($p < 1\mathrm{e}{-}4$), and with RoBERTa on SST-2, Slice Out underperforms Attention Mask by up to 4.72 points ($p < 1\mathrm{e}{-}4$). Marginalize is regularly more than 10 points worse than the best method. Overall, we recommend that, *when not using Counterfactual Training,* researchers use either the Attention Mask or Mask Token `Replace` functions.

**Counterfactual Training vs. Standard Training.** On the right side of Fig. 2, we see the effect of Counterfactual Training on model robustness for several `Replace` functions. We find that counterfactual inputs are much less OOD for Counterfactual-Trained models than for Standard models, regardless of the `Replace` function used. The improvement in robustness is up to **22.9** points. Moreover, the difference between `Replace` functions is almost entirely erased, though we do observe a statistically significant difference between Attention Mask and Zero Embedding with 80% of tokens removed (by 2.23 points, $p < 1\mathrm{e}{-}4$). Given these results, and following Sec. 4, **we ultimately recommend that researchers use Counterfactual Training with the Attention Mask, Mask Token, or Slice Out `Replace` function whenever they intend to create FI explanations.**

# 6 Explanation Methods and Experiments

## 6.1 Explanation Methods

We describe explanation methods we consider below, with implementation details in the Supplement.

**Salience Methods.** One family of approaches we consider assigns a scalar *salience* to each feature of an input. The key property of these scores is that they allow one to rank-order features. We obtain binarized explanations through selecting the top-$k$ features, or up to the top-$k$ features when some scores are negative (suggesting they should not be kept). We list the methods below:[4]

*1. LIME.* LIME estimates FI by learning a linear model of a model's predicted probabilities with samples drawn from a local region around an input [46]. Though it is common to use the Slice Out `Replace` function with LIME, we use the Attention Mask `Replace` function (following Sec. 5), meaning we sample local attention masks rather than local input sequences.

*2. Vanilla Gradients.* We obtain model gradients w.r.t. the model input as salience scores, one early method for explanation [33]. We compute the gradient of the predicted probability w.r.t. input token embeddings, and we obtain a single value per token by summing along the embedding dimension.

*3. Integrated Gradients.* We evaluate the Integrated Gradients (IG) method of Sundararajan et al. [57]. This method numerically integrates the gradients of a model output w.r.t. its input along a path between the observed input and a user-selected baseline. Given our results in Sec. 5, we use a repeated MASK token embedding for our baseline $\tilde{x}$ rather than the all-zero input suggested by Sundararajan et al. [57] for text models. We use the model's predicted probability as the output, and to obtain token-level salience scores, we sum the output of IG along the embedding dimension.

**Search Methods.** An alternative class of methods searches through the space of possible explanations. Search methods are regularly used to solve combinatorial optimization problems in machine learning [49, 5, 3, 16, 40]. All search methods use the Attention Mask `Replace` function, and the search space is restricted to explanations of the maximum allowable sparsity (or minimum, with Comprehensiveness), except for Anchors which takes a maximum number of features as a parameter.

*1. Random Search.* For each maximum explanation sparsity $k$ (or minimum, for Comprehensiveness), we randomly sample a set of $k$-sparse explanations, compute the current objective for each of them, and choose the best explanation under the objective.

*2. Anchors.* Ribeiro et al. [47] introduce a method for finding a feature subset that almost always yields the same model prediction as its original prediction for some data point, as measured among data points sampled from a distribution centered on the original data point. Explanations are also preferred to have high coverage, meaning the feature subset is likely to be contained in a local sample. The solution is identified via a Multi-Armed Bandit method combined with a beam search.

---

[4]In early experiments, we found that a parametric model (similar to [4, 2, 42]) performed far worse than other salience methods, and hence we leave out parametric models from further consideration.

*3. Exhaustive Search.* Exhaustive search returns the optimal solution after checking the entire solution space. This is prohibitively expensive with large inputs, as there is a combinatorial explosion in the number of possible explanations. However, we are able to exactly identify optimal explanations for short sequences, typically with 10 or fewer tokens.

*4. Gradient Search.* Fong and Vedaldi [19] propose to search through a continuous explanation space by gradient descent. We introduce a Gradient Search that uses discrete explanations, because continuous explanations do not reflect test-time conditions where discrete explanations must be used. For an input of length $L$, this method sequentially updates a continuous state vector $s \in \mathbb{R}^L$ via gradient descent to minimize a regularized cross-entropy loss between the model prediction on the input $x$ and the model prediction for `Replace`$(x, e_t)$, where $e_t$ is a discrete explanation sampled from a distribution $p(e_t | s_t)$ using the Gumbel-Softmax estimator for differentiability [37, 26]. The regularizer maintains sparsity of the solution. The final explanation is obtained from the last state $s_t$.

*5. Taylor Search.* Inspired by HotFlip [16], this method explores the solution space by forecasting the change in the objective using a first-order Taylor approximation. Specifically, this is a local search method with a heuristic function computed as follows: We first calculate the gradient $g \in \mathbb{R}^L$ of the cross-entropy loss (same loss as Gradient Search) with respect to the explanation $e_t$. Then we find the two indices $i$ and $j$ as the solution to $\arg \max_{i,j} \ g^{(i)} - g^{(j)}$. The next state is obtained by setting $e_t^{(i)} = 1$ and $e_t^{(j)} = 0$. This is a first-order approximation to the change in loss between the new and old state, based on the Taylor expansion of the loss [16]. Note when optimizing for Comprehensiveness, we use the $\arg \min$. Following Ebrahimi et al. [16], we ultimately use this search heuristic within a beam search, starting from a random explanation, with width $w = 3$.

*6. Ordered Search.* Next, we introduce a global search algorithm, Ordered Search. This method searches through all explanations in an order given by a scoring function $f : e \to \mathbb{R}$. We only require that $f$ is linear in $e$, as this allows for efficient ordering of the search space using a priority queue [3]. The algorithm proceeds by first estimating parameters for $f_\theta$, then searching through explanations in order of their score, $\theta^T e$. For the first stage, we obtain parameters for $f_\theta$ from the per-token salience scores given by LIME, which is the best salience method evaluated in Sec. 6. In the second stage, we enumerate the search space in order of the score given by $f_\theta$. We allocate 25% of the compute budget to the first stage and 75% to the second (measured in terms of forward passes).

*7. Parallel Local Search* (PLS). Lastly, we again consider the class of local search algorithms, which have a long history of success with constrained optimization problems [43, 3]. We propose a parallelized local search algorithm (PLS) tailored to the problem at hand. Given a number $r$ of parallel runs to perform, a proposal function `Propose`, and compute budget $b$ per run, an individual search proceeds as follows: (1) Sample a random initial explanation $e_1$ and compute the objective function for that explanation. (2) For the remaining budget of $b-1$ steps: sample a not-already-seen explanation $e_t$ according to `Propose` and adopt $e_t$ as the new state only if the objective is lower at $e_t$ than at the current state. The `Propose` function samples not-already-seen explanations from neighboring states to the current state. We use $r = 10$ parallel runs following tuning results.

## 6.2 Experimental Setup

**Data.** We compare the above explanation methods on six benchmark text classification datasets: SNLI [6], BoolQ [10], Evidence Inference [32], FEVER [58], MultiRC [29], and SST-2 [53]. One important distinction among these datasets is that BoolQ, FEVER, MultiRC, and Evidence Inference data points include both a *query* and an accompanying *document*. The query is typically critical information indicating how the model should process the document, and therefore we never replace query tokens. We use 500 test points from each dataset to compare methods. See Table **??** in the Supplement for dataset statistics, including average sequence length.

**Models.** We train ten seeds of BERT-Base models on each dataset [12], which we term Standard models. For each dataset we train another ten Counterfactual-Trained (CT) models using the Attention Mask `Replace` function, following the approach outlined in Sec. 4 (further details in Supplement).

**Controlling for Compute Budget.** We wish to control for the available compute budget in order to fairly compare explanation methods. Some explanations require a single forward and backward pass [51, 33], while others can require hundreds of forward and backward passes [57] or thousands of

Table 1: Explanation metrics across methods and datasets

| Dataset | Method | Sufficiency ↓ | | Comprehensiveness ↑ | |
|---|---|---|---|---|---|
| | | Standard Model | CT Model | Standard Model | CT Model |
| SNLI | LIME | 20.00 (2.02) | 27.08 (1.68) | 82.18 (2.82) | 75.34 (1.93) |
| | Int-Grad | 43.76 (3.27) | 32.91 (2.36) | 34.01 (2.55) | 43.22 (2.28) |
| | Anchors | 11.93 (1.53) | 30.96 (1.87) | 55.72 (2.60) | 48.86 (2.38) |
| | Gradient Search | 17.55 (1.47) | 33.98 (1.43) | 53.15 (2.53) | 49.36 (1.95) |
| | Taylor Search | 6.91 (1.10) | 28.00 (1.46) | 73.20 (2.57) | 66.76 (2.12) |
| | Ordered Search | -1.45 (0.93) | 15.06 (1.37) | 87.78 (2.41) | 84.67 (1.61) |
| | Random Search | -1.54 (0.96) | 15.38 (1.39) | 87.36 (2.47) | 84.63 (1.68) |
| | PLS | **-1.65** (1.07) | **14.16** (1.38) | **87.95** (2.55) | **86.18** (1.45) |
| BoolQ | LIME | 2.15 (1.75) | -1.56 (0.63) | 52.02 (3.69) | 36.25 (3.45) |
| | Int-Grad | 20.78 (3.57) | 9.05 (1.53) | 16.80 (1.57) | 12.20 (1.68) |
| | Anchors | 11.98 (2.62) | 6.07 (1.06) | 29.87 (4.17) | 15.46 (1.97) |
| | Gradient Search | 5.12 (1.41) | 1.65 (0.81) | 30.04 (2.58) | 17.65 (1.85) |
| | Taylor Search | 6.01 (1.33) | 2.28 (0.87) | 46.32 (3.89) | 26.65 (2.68) |
| | Ordered Search | 0.09 (0.84) | -2.58 (0.70) | 51.59 (3.52) | 34.36 (3.34) |
| | Random Search | -0.58 (0.63) | -2.51 (0.70) | 55.78 (3.71) | 31.62 (3.06) |
| | PLS | **-1.17** (0.47) | **-3.52** (0.88) | **72.78** (4.06) | **47.80** (3.57) |
| Evidence Inference | LIME | -16.07 (2.84) | -14.92 (1.38) | 47.60 (5.66) | 33.97 (4.22) |
| | Int-Grad | 1.22 (4.42) | -2.98 (1.68) | 26.51 (2.68) | 20.87 (2.57) |
| | Anchors | 7.08 (4.70) | 3.04 (0.99) | 25.01 (6.52) | 13.89 (1.55) |
| | Gradient Search | -10.57 (2.58) | -7.56 (1.46) | 31.73 (4.43) | 18.07 (2.13) |
| | Taylor Search | -4.55 (2.66) | -3.33 (1.27) | 41.95 (5.63) | 26.70 (3.00) |
| | Ordered Search | -16.80 (2.75) | -14.26 (1.36) | 45.37 (5.53) | 31.14 (3.73) |
| | Random Search | -17.05 (2.83) | -12.69 (1.30) | 42.81 (6.00) | 26.48 (3.15) |
| | PLS | **-20.76** (3.77) | **-20.33** (2.65) | **56.31** (9.81) | 38.71 (3.91) |
| FEVER | LIME | -0.24 (0.50) | 0.39 (0.96) | 33.86 (3.43) | 22.06 (2.36) |
| | Int-Grad | 9.72 (1.80) | 4.99 (1.40) | 17.81 (2.47) | 13.69 (1.71) |
| | Anchors | 6.19 (1.22) | 6.36 (1.10) | 20.82 (2.58) | 11.94 (1.84) |
| | Gradient Search | 0.66 (0.68) | 2.63 (1.12) | 19.26 (2.68) | 11.44 (1.65) |
| | Taylor Search | 4.17 (0.96) | 4.20 (1.20) | 24.51 (2.78) | 15.62 (1.85) |
| | Ordered Search | -1.26 (0.41) | -0.01 (0.90) | 31.79 (3.28) | 18.90 (2.46) |
| | Random Search | -1.51 (0.51) | -1.24 (2.33) | 32.47 (3.33) | 18.84 (2.11) |
| | PLS | **-2.04** (0.62) | **-3.66** (0.82) | **37.72** (3.28) | **24.07** (2.46) |
| MultiRC | LIME | -5.20 (1.18) | -5.90 (1.19) | 39.75 (4.84) | **28.57** (2.18) |
| | Int-Grad | 13.19 (3.14) | 4.66 (1.71) | 15.53 (3.39) | 11.84 (1.31) |
| | Anchors | 5.40 (3.34) | 3.33 (1.27) | 24.53 (8.77) | 14.55 (1.66) |
| | Gradient Search | -0.09 (1.33) | -0.73 (1.18) | 20.16 (2.92) | 11.41 (1.13) |
| | Taylor Search | 7.54 (2.53) | 1.43 (1.47) | 30.76 (4.04) | 20.15 (1.83) |
| | Ordered Search | -6.43 (0.98) | -5.49 (1.13) | 35.70 (4.40) | 24.38 (2.03) |
| | Random Search | -7.42 (1.08) | -5.97 (1.22) | 35.29 (4.59) | 22.19 (1.81) |
| | PLS | **-10.17** (1.43) | **-9.77** (1.49) | 39.95 (5.44) | 26.96 (2.19) |

forward passes [46]. Since this is expensive to perform, we limit each method to a fixed number of forward and backward passes (counted together), typically 1000 in total, to obtain a single explanation.

## 6.3 Main Results

In Table 1, we show Suff and Comp scores across methods and datasets, for both the Standard and Counterfactual-Trained (CT) models. 95% confidence intervals are shown in parentheses, obtained by bootstrap with data and model seeds resampled 100k times. Bolded numbers are the best in their group at a statistical significance threshold of $p < .05$. We give results for SST-2 in the Supplement, including for Exhaustive Search since we use short sequences there, as well as for Vanilla Gradient as it performs much worse than other methods. We summarize our key observations as follows:

**1. PLS performs the best out of all of the explanation methods, and it is the only method to consistently outperform Random Search**. Improvements in Sufficiency are statistically significant for every dataset using both Standard and CT models, with differences of up to 12.9 points over

LIME and 7.6 points over Random Search. For Comprehensiveness, PLS is the best method in 9 of 10 comparisons, 7 of which are statistically significant, and improvements are as large as 20.8 points over LIME and 17 points over Random Search.

**2. LIME is the best salience method on both Suff and Comp metrics, but it is still outperformed by Random Search on Sufficiency in 9 of 10 comparisons, by up to 21.5 points**. LIME does appear to perform better than Random Search on Comprehensiveness with three of five datasets for Standard models and four of five with CT models, where the largest improvement over Random Search is 7.49 points.

**3. Suff and Comp scores are often much worse for CT models than for Standard models**. With Random Search, for instance, Comp scores are worse for all datasets (by up to 24.16 points), and Suff scores are worse by 16.92 points for SNLI, though there are not large changes in Suff for other datasets. The differences here show that the OOD nature of counterfactual inputs can heavily influence metric scores, and they lend support to our argument about the OOD problem in Sec. 4. In particular, these metrics are more easily optimized when counterfactuals are OOD because it is easier to identify feature subsets that send the model confidence to 1 or 0.

Given the results above, we recommend that explanations be obtained using PLS for models trained with Counterfactual Training. Though explanation metrics are often worse for CT models, the only reason for choosing between Standard and CT models is that CT models' explanations are socially aligned, while Standard models' explanations are socially misaligned. It would be a mistake to prefer standardly trained models on the grounds that they are "more easily explained" when this difference is due to the way we unintentionally influence model behavior for OOD data points. When using CT models, however, we should be comfortable optimizing for Sufficiency and Comprehensiveness scores, and PLS produces the best explanations under these metrics.

We give additional results for RoBERTa and reduced search budgets in the Supplement.

# 7    Conclusion

In this paper, we provide a new argument for why it is problematic to use out-of-distribution counterfactual inputs when creating and evaluating feature importance explanations. We present our Counterfactual Training solution to this problem, and we recommend certain `Replace` functions over others. Lastly, we introduce a Parallel Local Search method (PLS) for finding explanations, which is the only method we evaluate that consistently outperforms random search.

# 8    Broader Impacts and Limitations

There are several positive broader impacts of improved feature importance estimation methods and solutions to the OOD problem. When model developers and end users wish to understand the role of certain features in model decisions, FI estimation methods help provide an answer. Access to FI explanations can allow for (1) developers to check that models are relying on the intended features when making decisions, and not unintended features, (2) developers to discover which features are useful to a model accomplishing a task, and (3) users to confirm that a decision was based on acceptable features or dispute the decision if it was not. Our solution to the OOD problem helps align FI explanations with the kind of information that developers and users expect them to convey, e.g. by limiting the influence of the model prior on the explanations.

Nevertheless, there are still some risks associated with the development of FI explanations, mainly involving potential misuse and over-reliance. FI explanations are *not* summaries of data points or justifications that a given decision was the "right" one. When explanations are good, they reflect what the model has learned, but it need not be the case that what the model has learned is good and worth basing decisions on. It is also important to emphasize that FI explanations are not perfect, as there is always possibly some loss of information by making explanations sparse. Trust in and acceptance of these explanations should be appropriately calibrated to the evidence we have for their faithfulness. Lastly, we note that we cannot guarantee that our Counterfactual Training will eliminate the influence of the random seed and model prior on explanations, meaning that FI explanations for the models we consider will still not be perfectly socially aligned. It will be valuable for future work to further explore how these factors influence the explanations we seek for models.

## Acknowledgements

We thank Serge Assaad for helpful discussion of the topics here, as well as Xiang Zhou, Prateek Yadav, and Jaemin Cho for feedback on this paper. We also want to thank our reviewers and area chair for their thorough consideration of this work. This work was supported by NSF-CAREER Award 1846185, DARPA Machine-Commonsense (MCS) Grant N66001-19-2-4031, Royster Society PhD Fellowship, Microsoft Investigator Fellowship, and Google and AWS cloud compute awards. The views contained in this article are those of the authors and not of the funding agency.

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
