# Supplement to "The Out-of-Distribution Problem in Explainability and Search Methods for Feature Importance Explanations"

## A   Method Implementation and Hyperparameter Tuning Details

### A.1   `Replace` **Functions**

1. **Attention Mask.** To make this a differentiable function, we compute the function by taking the element-wise product between the attention distribution and the binary attention mask, then renormalizing the attention probabilities to sum to 1. The difference between this approach and deleting a token from an input text is that positional embeddings for retained tokens are unchanged.

2. **Marginalize.** As in Kim et al. [31], we use a pretrained BERT-Base as our generative model (or a RoBERTa-Base model, when the classifier is a RoBERTa model). The final prediction is obtained from the marginal log probability as $\arg\max_y \ln \sum_{\tilde{x} \sim p_\phi(\tilde{x}|x,e)} p_\theta(y|\tilde{x})p_\phi(\tilde{x}|x,e)$, where $p_\phi(\tilde{x}|x,e)$ is the distribution over imputed tokens. Since computing this marginal distribution is quite expensive, we adopt a Monte Carlo approximation common to past work [31, 69]. Using a subset of SNLI validation data, we tune the number of samples over sizes in $\{10, 25, 50, 100\}$, selecting for maximum robustness. Surprisingly, 10 samples performed the best in terms of robustness, though the margin was small over the other values. Consequently, we select a value of 10, which also allows us to evaluate this method at scale due to its relative computational efficiency. This finding is similar to the results in Yi et al. [69], who ultimately use a value of 8 samples for Monte Carlo estimation of the marginal distribution. This method is still over ten times slower than other `Replace` functions given the need to perform many MLM forward passes.

3. **MASK Token**. Described in main paper.

4. **Slice Out**. Desribed in main paper.

5. **Zero Vector**. Described in main paper.

### A.2   Explanation Methods

**LIME.**  For a data point $x$, we train a model $m_\phi$ minimizing an MSE weighted by the kernel $\pi$ and regularized by $\Omega$,

$$\sum_{i=1}^{N} \pi(x, \tilde{x}_i)(m_\phi(\tilde{x}_i) - f_\theta(\tilde{x}_i)_{\hat{y}})^2 + \Omega(\phi)$$

where $f_\theta(x)_{\hat{y}}$ is the task model's predicted probability, local samples $\tilde{x}_i$ have attention masks that are imputed with a random number of 0s, and $\Omega$ is the default `auto` regularization in the LIME package.

We next specify the form of the weight function $\pi$, the regularization method $\Omega$, and the distribution of perturbed data points $p(\tilde{x}|x_i)$, which are all set to the default LIME package settings. The weight function $\pi$ is an exponential kernel on the negative cosine similarity between data points multiplied by 100. The perturbation distribution is over binary vectors: in every sample, a uniformly random number of randomly located elements are set to 0, and the remainder are kept as 1. Lastly, $\Omega$ is to perform forward selection when there are no more than 6 features (i.e. perform greedy local search in the space of possible feature sets, starting with no features and adding one feature at a time). When there are more than six features, ridge regression is used, then the top $k$ features according to the product of their feature weight and the observed feature value (0 or 1 in our case). We use the regression weights as the final salience scores.

**Vanilla Gradient.**  We obtain model gradients w.r.t. the model input as salience scores, one early method for explanation [34]. We compute the gradient of the predicted probability w.r.t. input token embeddings, and we obtain a single value per token by summing along the embedding dimension.

**Integrated Gradients.**  The salience for an input $x$ with baseline $\tilde{x}$ is given as

$$(x - \tilde{x}) \times \int_{\alpha=0}^{1} \frac{\partial f(\tilde{x} + \alpha(x - \tilde{x}))}{\partial x} d\alpha.$$

We use the input embeddings of a sequence as $x$. By the Completeness property of IG, token-level salience scores still sum to the difference in predicted probabilities between the observed input and the baseline.

**Random Search.**   Using a subset of SNLI validation points, we tune this method over two possible search spaces: the space of all $k$-sparse explanations, when the sparsity levels allows up to $k$ tokens to be retained (or no lower than $k$ tokens, for Comprehensiveness), and the space of all allowably sparse explanations. We find it preferable to restrict the search space to exactly $k$-sparse explanations. We adopt this same search space for all other search methods.

**Anchors.**   We use the `anchor-exp` package made available by Ribeiro et al. [50] for our experiments, with two modifications. First, we limit the compute budget used in this method to 1000 forward passes (as with all search methods). Second, thoughwe sample locally around inputs using the default argument `masking_str='UNKWORDZ'`, we use the Attention Mask `Replace` function for computing the model forward pass, as we do with all search methods. Besides this, we call `explain_instance` with default parameters, and we refer the reader to Ribeiro et al. [50] for additional details. Note that we distinguish results on Sufficiency and Comprehensiveness in terms of the maximum number of features selected by the explanation. Additionally, this method has over a 3x slower wall-clock runtime compared to our search methods used with the same compute budget (in terms of model forward passes), and as a result we are constrained to reporting results across a smaller number of model seeds for each dataset (between 3 and 10, rather than always 10).

**Gradient Search.**   For an input of length $L$, this method sequentially updates a continuous state vector $s \in \mathbb{R}^L$ via gradient descent in order to minimize a regularized cross-entropy loss between the original model prediction $\hat{y}$ and the predicted probability given the input $\tilde{x} = \texttt{Replace}(x, e_t)$. The explanation $e_t$ is sampled as follows: $e^{(d)} \sim$ Gumbel-Softmax$(s^{(d)})$, for $d = 1 : L$. The new state is $s_{t+1} \leftarrow s_t - \alpha \nabla_s \mathcal{L}(\hat{y}, f(\tilde{x})_{\hat{y}})$, though note that we use an AdamW optimizer for this step. By virtue of the differentiable Attention Mask `Replace` function and the Gumbel-Softmax estimator [39, 27], this loss is differentiable w.r.t. $s$. The regularizer is an $\ell_2$ penalty on the difference between the expected sparsity $\sum_{d=1}^{L} \sigma(s^{(d)})$ and a target sparsity, set to ceiling$(.05 \cdot L)$, which is designed to encourage searching through sparse explanations. The final salience scores are given by $s$, with the probabilistic interpretation that $\sigma(s_j)$ is the probability that token $j$ is in the optimal explanation.

We observe that this search method is equivalent to fitting a non-parametric model to the dataset with the objective $\mathcal{L}$ given above. Recently, many parametric models have been proposed for sampling explanations for individual data points [5, 3, 66, 45, 9, 12]. In early experiments, we found that a parametric model performed far worse than this non-parametric approach, and hence we leave out parametric models from further consideration. This is perhaps unsurprising given how hard it may be to learn a map from inputs to good explanations for all data.

Now we give more details to checkpoint selection, weight initialization, regularization, and tuning for Gradient Search. For checkpoint selection: we select the search state that achieves the best Sufficiency (or Comprehensiveness) as measured once every $m$ gradient updates. We do so because checking these metrics consumes some of the available compute budget (see Supplement B.3), and therefore we check the metric value at intervals for purposes of checkpointing. In our experiments, we check the metric every 20 gradient updates and search until the total budget has been consumed. For initialization: a random initial starting point is sampled from a Multivariate Normal distribution centered around 0, with $\Sigma = I$. For regularization and other tuning details, we perform sequential line searches over hyperparameters, according to Sufficiency scores on a subset of BoolQ data points. To tune a specific hyperparameter, we set all other hyperparameters to some default values. We refer to the hyperparameters we use after tuning as "final" hyperparameters, which are listed in the table below (note: Number of Samples is the number of sampled explanations per gradient update).

| Hyperparameter | Default | Final | Range |
|---|---|---|---|
| Number of Samples | 10 | 1 | 1, 10, 20, 40 |
| Optimizer | AdamW | AdamW | AdamW, SGD |
| Scheduler | None | None | None, Linear, Step, Cosine |
| Learning Rate | 0.2 | 0.1 | 0.01, 0.05, 0.1, 0.2, 0.4 |
| Sparsity Weight | 1e-3 | 1e-3 | 1e-1, 5e-2, 1e-2, 5e-3, 1e-3, 5e-4, 0 |
| Target Sparsity | 0.1 | 0.05 | 0.03, 0.05, 0.1, 0.2, 0.3, 0.4 |

**Taylor Search.** At time step $t$, the state is an explanation $e_t$, and a heuristic is evaluated on neighboring states in order to select the next state to compute the objective on. The search space is all $k$-sparse explanations, and therefore neighboring states are those with Hamming distance 2 to the current state (with one retained token being hidden and one hidden token being retained). The heuristic is the projected change in the cross-entropy loss between the model's original prediction $\hat{y}$ and this label's probability given the input Replace$(x, e)$, for a proposed explanation $e$, which is computed as such: We first calculate the gradient $g \in \mathbb{R}^L$ of the cross-entropy loss with respect to the explanation $e_t$, which is possible with the differentiable Attention Mask Replace function. Then, when optimizing for Sufficiency, we find the two indices $i$ and $j$ as the solution to $\arg\max_{i,j} g^{(i)} - g^{(j)}$ such that the sparsity is maintained by flipping both tokens, meaning $e^{(i)} = 1$ ($x^{(i)}$ is retained) and $e^{(j)} = 0$ ($x^{(j)}$ is hidden). The next state is obtained by setting $e_t^{(i)}$ and $e_t^{(j)}$ to these values. This is a first-order approximation to the change in loss between the new and old state, based on the Taylor expansion of the loss [17]. Note when optimizing for Comprehensiveness, we use the $\arg\min$. Following Ebrahimi et al. [17], we ultimately use this search heuristic within a beam search, starting from a random explanation, with width $w = 3$.

Hyperparameters for Taylor Search are listed below. We performed tuning with Taylor Search for Sufficiency on a subset of BoolQ validation points, and ultimately we selected the largest, best performing pair of values given the available compute budget.

| Hyperparameter | Default | Final | Range |
|---|---|---|---|
| Beam Width | 2 | 5 | 1,2,3,4,5 |
| Number of Steps | 50 | 50 | 50, 100, 200 |

**Ordered Search.** More complicated forms for $f$, such as being quadratic in $e$, would make finding the optimum of the function computationally intractable, let alone a full rank-ordering of solutions [4].

Using Sufficiency scores on a subset of SNLI validation data, we tune over the ratio between compute budget used in estimating the model $f_\theta$ (i.e. salience scores) and the budget used for the search. Out of 1000 steps, we consider using up to $m$ steps for estimating the salience scores via LIME, where $m \in \{10, 100, 200, 250, 500, 750\}$, ultimately using $m = 250$.

**Parallel Local Search.** Here we specify the Propose function used in Parallel Local Search, and we describe some additional implementation details, some comparisons we performed with Simulated Annealing, and tuning for the number of parallel searches $r$. The Propose function samples new explanations by starting a random walk from the current explanation that ends when a not-before-seen explanation is encountered. As in Taylor Search, neighboring states have Hamming distance 2. Though we parallelize this search method, we maintain a shared set of previously-seen explanations and compute the Propose function serially at each step so that we never compute the more expensive objective function on the same explanations.

A similar algorithm, Simulated Annealing, uses a probabilistic update condition that favors exploration early on in the search and exploitation later in the search [4]. We find it preferable to use a deterministic update rule, moving to the new state if and only if its objective value is better than the old state. Lastly, following tuning results, we use $r = 10$ parallel runs for all experiments, meaning that each run has a budget $b = 100$ when the overall method budget is 1000 forward passes. The value of $r$ is tuned over the set $\{1, 5, 10, 25\}$. We note that using a value greater than 1 significantly improves the wall-clock runtime of this algorithm, as the batched forward passes performed when $r$ searches are

Table 2: Dataset statistics

| Dataset | # Classes | Split | Size | Avg. document length | Avg. query length |
|---------|-----------|-------|------|---------------------|-------------------|
| SNLI | 3 | Train | 5000 | 24.8 | - |
|  |  | Validation | 9823 | 24.4 | - |
|  |  | Test | 9807 | 24.4 | - |
| BoolQ | 2 | Train | 9427 | 121.9 | 9.4 |
|  |  | Validation | 3270 | 119.8 | 9.3 |
| FEVER | 2 | Train | 97957 | 342.3 | 10.4 |
|  |  | Validation | 6122 | 291.2 | 10.7 |
|  |  | Test | 6111 | 278.7 | 10.7 |
| Evidence Inference | 3 | Train | 7958 | 483.1 | 25.3 |
|  |  | Validation | 972 | 484.4 | 23.6 |
|  |  | Test | 959 | 482.0 | 27.0 |
| SST-2 2 | 2 | Train | 67349 | 11.3 | - |
|  |  | Validation | 872 | 23.2 | - |
|  |  | Test | 1821 | 10.8* | - |
| MultiRC 2 | 2 | Train | 24029 | 326.9 | 21.2 |
|  |  | Validation | 3214 | 326.1 | 20.7 |
|  |  | Test | 4848 | 314.8 | 20.8 |

done in parallel are much more efficient than performing a greater number of forward passes with only 1 input.

## A.3 Model Training Details and Experiment Runtimes

We now give implementation details for training models on our six datasets. The models include BERT-Base or RoBERTa-Base models drawn from the Hugging Face Transformers library [68], trained with AdamW [37] using a learning rate of 1e-5, and in general we select the best model training checkpoint according to the validation set accuracy. When training models for our analysis of counterfactual OOD-ness in Sec. 5, we train Standard models for 20 epochs and Counterfactual-Trained models for 10 epochs, since in the latter case we effectively double the number of inputs per batch. For our explanation evaluation experiments in Sec. 6, we train all models for 10 epochs. Note that in every experiment we train ten models using ten different random seeds.

All experiments are conducted on a NVIDIA RTX 2080 GPU. Wall-clock runtimes for training models are: for Sec. 5 experiments, .6 hours per SNLI model and 1 hour per SST-2 model; for Sec. 6, training one model takes 4.8 hours for FEVER, .7 hours for BoolQ, 1.5 hours for SNLI, 2.9 hours for MultiRC, 1.7 for Evidence Inference, and 3.9 hours for SST-2.

For analysis and explanation evaluation experiments, we report the following runtimes: Sec. 5 experiments take up to 12 hours for robustness analysis for each `Replace` function (across seeds, models, and datasets), except for Marginalize which takes close to 48 hours. We give max runtimes across datasets for obtaining and evaluating explanations to provide an upper bound on wall-clock runtime given the variable sequence lengths between datasets, meaning we report runtimes for the Evidence Inference dataset. With a compute budget of 1000 forward/backward passes, we find observe that obtaining a set of explanations at one sparsity level using 500 points for one BERT-Base model takes 2.5 hours for LIME, 2.5 minutes for Vanilla Gradient, 11 hours for Integrated Gradients, 10 hours for Gradient Search, 17 hours for Anchors, 5 hours for Taylor Search, 3 hours for Ordered Search, 5 hours for Random Search, and 5 hours for Parallel Local Search.

# B  Experimental Details

## B.1  Data Preprocessing

We use datasets as prepared by the ERASER dataset suite [14], which are publicly available and distributed under the original licenses for each dataset (including Creative Commons and MIT License),[5] as well as SST-2 which is publicly available under a Creative Commons license [58].[6] Note that BoolQ has only one split for evaluation. Additionally, note that for experiments in Sec. 6, we use a subset of 50k points for training models on SNLI, and we use the 500 shortest SST-2 test points according to the number of tokens given by the BERT tokenizer [67] in order to compare with exhaustive search for this dataset.

For preprocessing, input text is split by spaces into a list of words. We tokenize each word independently and concatenate the resulting tokens. Each input consists of a document section and a query section. The document section contains the document while the query section contains the question. The two sections are separated by [SEP]. The entire input is preceded by a [CLS] token and followed by a [SEP] token. For inputs longer than 512 tokens (the maximum input length for our task model Bert), we truncate the input document so that the entire input is shorter or equal to 512 tokens.

## B.2  Analysis of Counterfactual Input OOD-ness Details

In this evaluation, the tokens to be hidden from the model are selected uniformly at random without replacement. We sample 10 random masks per data point and take the majority prediction on the corresponding 10 ablated data points as the model's overall prediction. The exception to this procedure is Marginalize, since it is much more computationally expensive than other methods, and therefore we use only one random explanation per input. Note that we use a subset of 10k train points for each task, and we perform this experiment on validation splits because the experiment motivates method design choices in Sec. 6.1.

## B.3  Compute Budget Details

In this section we describe how the compute budget is spent by each explanation method. Note that we consider both creating and evaluating explanations to draw from the available compute budget, because some methods compute the Suff and Comp metrics while obtaining an explanation, whereas others leave these metrics to be checked after an explanation is settled on. We describe the standard case in this paper of 1000 forward and backward passes per final metric value.

1. LIME uses 996 forward passes to obtain an explanation, then 4 forward passes to obtain a final metric value (one per sparsity level).
2. Vanilla Gradient uses only a single forward and backward passes. This is our only method that uses a fixed compute budget.
3. Integrated Gradients uses 498 forward and backward passes and 4 forward passes to obtain the final metric value.
4. Taylor Search uses no more than 1000 forward passes, given the beam width and number of steps described in Supplement A.
5. The remaining search methods (including Ordered Search, Random Search, Parallel Local Search) all use 1000 forward passes in total, since these methods involve exactly computing the objective value at each step, so the metrics do not need to be recomputed after explanations are obtained.

# C  Additional Results

**SST-2 Results.** Here we discuss results for SST-2, which are shown in Table 4. Our primary observation here is that most of the search methods we consider perform as well as Exhaustive Search (for sequences short enough to exhaustively search). The closest to Exhaustive Search is Parallel Local Search, which exactly identifies the optimal explanation for the Sufficiency metric and comes

---

[5]http://www.eraserbenchmark.com/
[6]https://www.kaggle.com/atulanandjha/stanford-sentiment-treebank-v2-sst2

Table 3: Model accuracies

| Dataset | Standard Acc. | CT Acc. |
|---|---|---|
| SNLI | 85.84 (0.69) | 85.08 (0.71) |
| BoolQ | 74.16 (1.62) | 73.76 (1.62) |
| FEVER | 89.66 (0.76) | 89.72 (0.76) |
| Evidence Inference | 58.81 (3.12) | 57.35 (3.13) |
| SST-2 | 92.89 (1.18) | 92.43 (1.21) |
| MultiRC | 68.96 (1.30) | 67.76 (1.32) |

Table 4: Explanation metrics for SST-2

| Dataset | Method | Sufficiency ↓ | | Comprehensiveness ↑ | |
|---|---|---|---|---|---|
| | | Standard Model | CT Model | Standard Model | CT Model |
| | LIME | 1.98 (0.84) | 5.92 (0.93) | 52.42 (2.92) | 45.75 (2.49) |
| | Anchors | 3.44 (0.96) | 17.69 (1.64) | 30.03 (3.13) | 24.19 (2.54) |
| | Taylor Search | 0.09 (0.50) | 5.02 (0.79) | 45.65 (3.11) | 38.91 (2.70) |
| SST-2 | Ordered Search | -0.91 (0.47) | 2.69 (0.79) | 56.24 (2.82) | 49.21 (2.48) |
| | Random Search | -0.91 (0.48) | 2.70 (0.79) | 56.11 (2.85) | 48.98 (2.49) |
| | PLS | -0.91 (0.51) | 2.68 (0.85) | 56.28 (2.84) | 49.25 (2.53) |
| | Exhaustive Search | -0.91 (0.51) | 2.68 (0.85) | 56.29 (2.84) | 49.26 (2.53) |

within .01 of the optimal Comprehensiveness value. Meanwhile, the best salience method (LIME) underperforms these search methods by between 2.86 and 3.86 points, showing that salience methods fall well behind search methods in this scenario.

**Search Method Performance Over Time.** In Figure 3, we show search performance across time steps for the three best performing search methods, Random, Ordered, and Parallel Local. Note that Ordered Search begins at step 251 since the first 250 forward passes are allocated to computing LIME, and Parallel Local Search begins at step 10 since we use 10 parallel runs. We see that Parallel Local outperforms Random early on and then continues to remain the preferable method, as differences at step 1000 are statistically significant at $p < .05$. In fact, for FEVER, where the search space is larger, Parallel Local will clearly continue to improve with additional compute, while Random and Ordered plateau in performance by 1000 steps.

Each of these search methods tends to achieve good performance even by 250 steps. We report results using this **reduced search budget** across datasets in Table 5 in the Supplement. We find that search methods outperform salience methods in 15 of 24 comparisons (with 8 favoring LIME), suggesting that search methods can outperform salience methods even with a far smaller compute budget. This is relevant if, for instance, one needs to obtain explanations at multiple sparsity levels with a single compute budget for all of the explanations, which could be useful for applications requiring all explanations to be visualized at once (or visualized on demand with no latency).

**Counterfactual-Trained Model Accuracy.** In Table 3, we show model accuracies on test sets for standardly trained models (Standard) and Counterfactual-Trained (CT) models, with 95% confidence intervals reflecting sample variance in parentheses. These accuracies are from the single best performing seed for each dataset, according to dev set accuracies, out of 10 trained models. Note that for SST-2, we report dev set numbers as test labels are unavailable.

We observe that differences in accuracies are typically less than 1 point between the two training algorithms. On average across the datasets, the difference is about 0.7 points. This is a small but consistent gap, and hence it will be valuable for future work to study the causes of this gap and identify ways to align the train and explanation-time distributions without losing any model accuracy on test data.

**Results with Reduced Search Budget.** In Table 5, we give results for a reduced search budget of 1000 forward passes *across* sparsity levels, i.e. 250 per sparsity level. This setup favors salience-based methods which can easily give explanations at varying sparsity levels. With too many sparsity levels, **this evaluation is heavily biased toward salience-based methods**, since it spreads the compute for search methods across sparsity levels. We use a constant budget per sparsity for results in the

Figure 3: Search method performance over time on FEVER and SNLI with Counterfactual-Trained models, for searches that ran for 1000 forward passes. Shaded areas represent 95% confidence intervals accounting for seed variance using 10 random seeds.

main paper because we view the use of multiple sparsity levels as an attempt to average results across possible desired settings, and explanations at multiple sparsities may not always be needed. But ultimately, user preferences will dictate whether explanations at multiple levels of sparsity are desired. This discussion aside, the results are as follows: compared to the best salience-based method, LIME, **search methods are preferable in 15 of 24 comparisons**, while **LIME is preferable in 8 of 24 comparisons** (at statistical significance of $p < .05$). Comparing within search methods, PLS is preferable to Random Search 7 times and Random Search is preferable one time (at $p < .05$). We observe that LIME performs best on Comprehensiveness, and therefore we suggest that LIME may be the best method when explanations are desired at many sparsity levels, the compute budget is heavily limited, and one is optimizing for Comprehensiveness. Otherwise, PLS remains the preferable method.

**Results with RoBERTa Models.** Shown in Table 6, we give a comparison between LIME, Random Search, and PLS using RoBERTa-Base as our task model, as opposed to the BERT-Base setting in the main paper experiments. Results are given for a subset of datasets, with the same compute budget as in the main paper, using 10 Counterfactual-Trained RoBERTa-Base models on each dataset. We observe similar trends and improvements using PLS as with BERT-Base. PLS is the best method in each condition and consistently outperforms Random Search, unlike LIME.

**Weight of Evidence Metrics.** Note that we can also measure the Sufficiency and Comprehensiveness metrics in terms of a difference in log-odds (i.e. weight of evidence) rather than probabilities, which reflects differences in evidence for classes before this evidence is compressed to the bounded probability scale [51, 1]. In this case, Sufficiency e.g. is computed as

$$\text{Suff}_{\text{WoE}}(f_\theta, x, e) = \ln \frac{f_\theta(x)_{\hat{y}}}{f_\theta(x)_{\neg \hat{y}}} - \ln \frac{f_\theta(\text{Replace}(x, e))_{\hat{y}}}{f_\theta(\text{Replace}(x, e))_{\neg \hat{y}}} \tag{2}$$

where $f_\theta(x)_{\neg \hat{y}}$ is the sum of all class probabilities except the predicted class, meaning each term is the log-odds in favor of $\hat{y}$. In Sec. 6 we describe results for the standard difference-in-probabilities version of each metric as well as the weight-of-evidence versions.

In Table 7, we compare the default difference-in-probability version of our metric (reported in the main paper) with the weight-of-evidence version, which is used by [51, 1]. We do not observe any notable differences in the trends between the metrics. We report one case where a hypothesis test is statistically significant with WoE but not the difference in probabilities; however, the difference between $p$-values is negligible ($p = .052$ vs. $p = .030$).

## D    Discussion

**Should We Prefer Counterfactual-Trained Models If They Are Harder to Explain?** This question is raised by the fact that Suff and Comp scores are often worse for CT models (see Sec. 6).

Table 5: Explanations metrics with a reduced search budget

| Dataset | Method | Sufficiency ↓ | | Comprehensiveness ↑ | |
|---|---|---|---|---|---|
| | | Standard Model | CT Model | Standard Model | CT Model |
| SNLI | Vanilla Grad | 59.41 (2.42) | 63.41 (0.81) | 7.08 (1.63) | 5.84 (1.06) |
| | LIME (1000) | 20.00 (2.02) | 27.09 (1.68) | 82.17 (2.82) | 75.34 (1.94) |
| | Ordered Search (250) | -1.19 (0.87) | 16.23 (1.45) | 87.01 (2.40) | 83.31 (1.73) |
| | Random Search (250) | -1.34 (0.90) | 16.82 (1.46) | 86.10 (2.52) | 82.45 (1.82) |
| | PL Search (250) | **-1.50** (0.96) | **15.30** (1.37) | **87.25** (2.42) | **84.57** (1.68) |
| BoolQ | Vanilla Grad | 30.81 (3.44) | 16.40 (2.13) | 2.43 (0.70) | 2.25 (0.73) |
| | LIME (1000) | 2.14 (1.75) | -1.56 (0.64) | 52.03 (3.67) | **36.26** (3.44) |
| | Ordered Search (250) | 1.44 (1.29) | -1.71 (0.70) | 43.33 (3.18) | 27.78 (3.00) |
| | Random Search (250) | 0.09 (0.82) | -1.84 (0.66) | 49.46 (3.55) | 27.58 (2.74) |
| | PL Search (250) | **-0.56** (0.57) | **-2.61** (0.68) | **54.85** (3.78) | 31.98 (2.97) |
| Evidence Inference | Vanilla Grad | 20.76 (4.14) | 12.96 (2.13) | 2.92 (1.31) | 1.57 (0.56) |
| | LIME (1000) | -16.07 (2.84) | -14.93 (1.38) | **47.61** (5.66) | **33.97** (4.19) |
| | Ordered Search (250) | -14.47 (2.65) | -11.67 (1.34) | 39.69 (4.83) | 26.51 (3.15) |
| | Random Search (250) | -15.28 (2.67) | -10.86 (1.28) | 38.79 (5.46) | 23.65 (2.80) |
| | PL Search (250) | **-16.18** (3.14) | **-13.78** (2.53) | 41.90 (8.96) | 25.27 (2.78) |
| FEVER | Vanilla Grad | 19.63 (2.39) | 13.21 (1.81) | 1.52 (0.60) | 1.02 (0.41) |
| | LIME (1000) | -0.24 (0.50) | 1.36 (2.13) | **33.86** (3.44) | **22.06** (4.10) |
| | Ordered Search (250) | -0.73 (0.40) | 1.00 (0.90) | 28.70 (3.18) | 16.30 (2.26) |
| | Random Search (250) | **-1.16** (0.50) | -0.30 (2.07) | 29.13 (3.18) | 16.58 (1.91) |
| | PL Search (250) | -1.06 (0.44) | **-0.36** (2.04) | 25.81 (2.87) | 15.22 (1.84) |
| MultiRC | Vanilla Grad | 21.16 (3.47) | 11.80 (1.38) | 3.75 (1.18) | 1.74 (0.79) |
| | LIME (1000) | -5.20 (1.19) | **-5.91** (1.19) | **39.75** (4.80) | **28.57** (2.18) |
| | Ordered Search (250) | -5.02 (1.03) | -4.16 (1.10) | 33.26 (4.25) | 21.95 (1.92) |
| | Random Search (250) | **-6.08** (1.17) | -4.86 (1.20) | 32.31 (4.30) | 19.95 (1.67) |
| | PL Search (250) | -5.20 (1.30) | -4.91 (1.27) | 28.38 (3.85) | 17.46 (1.49) |
| SST-2 | Vanilla Grad | 47.26 (3.79) | 46.83 (1.41) | 2.56 (0.71) | 3.01 (1.00) |
| | LIME (1000) | 1.97 (0.84) | 5.92 (0.93) | 52.42 (2.93) | 45.74 (2.52) |
| | Ordered Search (250) | -0.91 (0.47) | 2.71 (0.79) | 55.90 (2.84) | 48.96 (2.50) |
| | Random Search (250) | -0.91 (0.46) | 2.73 (0.75) | 55.44 (2.52) | 48.31 (2.18) |
| | PL Search (250) | -0.91 (0.47) | 2.69 (0.79) | 56.14 (2.84) | 49.08 (2.50) |

Table 6: Explanation metrics for RoBERTa-Base models

| Dataset | Method | Sufficiency ↓ | Comprehensiveness ↑ |
|---|---|---|---|
| SNLI | LIME | 24.42 (1.65) | 71.30 (2.63) |
| | Random Search | 11.52 (1.46) | 81.43 (2.59) |
| | PLS | **9.73** (1.41) | **83.44** (2.31) |
| FEVER | LIME | 1.22 (0.81) | 25.09 (2.56) |
| | Random Search | 1.10 (0.78) | 22.46 (2.50) |
| | PLS | **-0.40** (0.63) | **27.61** (2.68) |

We suggest that the only reason for choosing between Standard and CT models is that CT models' explanations are not influenced by the model prior and random seed to the extent that Standard models' explanations are, as we argued in Sec. 4. If one prefers explanations (and explanation metrics) to reflect what a model *has learned from the data*, rather than the model prior and random seed, one would prefer to use CT models. It would be a mistake to prefer the standardly trained models on the grounds that they are "more easily explained" when this difference is due to the way we unintentionally influence model behavior for out-of-distribution data points.

**In Defense of Searching For Explanations.** De Cao et al. [12] argue against using a certain kind of search to find feature importance explanations on the grounds that it leads to "over-aggressive pruning" of features that a model does in fact rely on for its output. In their case, the objective of the search method is to find "a subset of features (e.g. input tokens) ... [that] can be removed without affecting the model prediction." They suggest that this method is susceptible to *hindsight*

Table 7: Explanation metrics with weight of evidence

| Dataset | Method | Sufficiency ↓ | | Comprehensiveness ↑ | |
| --- | --- | --- | --- | --- | --- |
| | | Diff. in Probs | WoE | Diff. in Probs | WoE |
| Evidence Inference | LIME | -14.93 (1.38) | -0.98 (0.14) | 33.97 (4.17) | 1.76 (0.29) |
| | Random Search | -12.71 (1.29) | -0.81 (0.13) | 26.50 (3.15) | 1.35 (0.21) |
| | PLS | **-20.33** (2.46) | **-1.44** (0.14) | 38.71 (3.52) | **2.09** (0.25) |
| MultiRC | LIME | -5.90 (1.19) | -0.33 (0.09) | **28.57** (2.18) | **1.54** (0.17) |
| | Random Search | -6.58 (1.20) | -0.39 (0.09) | 22.79 (1.78) | 1.24 (0.13) |
| | PLS | **-9.77** (1.45) | **-0.63** (0.13) | 26.96 (2.05) | 1.45 (0.16) |

*bias*, asserting that "the fact that a feature can be dropped does not mean that the model 'knows' that it can be dropped and that the feature is not used by the model when processing the example." They provide the example of a model that (successfully) counts whether there are more 8s than 1s in a sequence, where they take issue with the fact that the search method would return a single 8 as its explanation for any sequence with more 8s than 1s, since this preserves the model prediction of "more 8s than 1s." The problem with this explanation, it is said, is that it removes digits that are most certainly being counted by the model when it comes to its decision. They provide empirical evidence that a learned model does in fact count all 8s and 1s before deciding which are more numerous.

One response to this argument is that if one obtains the optimal solution to an optimization problem and is not satisfied with it, then the objective must not be capturing everything that we care about, and the issue is not with the optimization method (i.e. search) that is employed. In the case at hand, we should first note that the objective is actually under-specified. De Cao et al. [12] suppose the search method returns the *maximal set* of tokens that can be removed while maintaining the model prediction, but the objective is not given with any preference for explanation *sparsity* (only that the removed tokens are a "subset" of the input). However, De Cao et al. [12] would take issue with search-based explanations regardless of whether the search method returns the minimal or maximal subset of tokens that can be removed without changing the model prediction. This is because they want the explanation to identify tokens that are "used by the model when processing the example." This criterion is not formalized, but the problem must be that it is a different criterion than the search objective, which is to find a feature subset that preserves the model prediction. After formalizing the notion of a feature being "used by a model," one should then be able to search for good explanations under the corresponding objective.

**Explanation Distribution at Train Time.** We reiterate that, to exactly match train and test-time distributions, models would be trained on counterfactual inputs drawn from explanation methods themselves, rather than simply random explanations. For now, this remains prohibitively expensive, as it would increase the number of forward passes required during training by up to 1000x depending on the budget to be used when explaining predictions. We explored methods based on training on the true counterfactual distribution (i.e., based on real explanations) to a limited extent, such as using real explanations only in the last training epoch. However, this alerted us to a few obstacles in such approaches: (1) this training distribution is non-stationary, as the FI explanations will change potentially with each gradient update, (2) these experiments were still quite expensive and required selecting a specific model checkpoint as the final model, which might not perform as well on accuracy metrics, and (3) we found that the Suff and Comp metrics were sometimes similar to Standard models, suggesting that these models were ultimately not as robust to the counterfactuals as the CT models were (see Conclusion 3 in Sec. 6.3). Future work in reducing the costs of obtaining explanations will help set the stage for more closely aligning the train and test time explanation distributions. In this regard, there may be applicable insights to draw from work on efficiently improving model robustness to local perturbations, such as Miyato et al. [41].

Another question that arises during training is whether applying the `Replace` function to $x_i$ implies that the label $y_i$ should change. It may *seem* problematic, for example, to fit a model to (`Replace`$(x, e)$, $y$) pairs if removing even a small number of tokens in $x$ tends to flip the label. However, we note that what the model sees is an input with, e.g., MASK tokens in place of some tokens, and what an optimal model will do in this situation is make a prediction based on what evidence is available to it, with the knowledge that some features that could have influenced the label have been removed from the input. That is, based on the overall data distribution, such a model should produce appropriately

calibrated outputs for inputs where most of the visible evidence suggests the labels to be $y_1$, while if the removed evidence had been visible, the label could be seen to be $y_2$.

**Measuring Compute Budget.** While we use the number of forward and backward passes as our compute budget for each method, wall-clock time will continue to be a useful and practical measure of compute for comparing methods. We note that batching forward passes on a GPU will significantly speed up method runtimes for a single data point while keeping the number of forward passes constant. In our experiments, this means that PLS is very efficient compared to Gradient Search, which does not batch inputs in the forward or backward pass. We use forward and backward passes as the unit of our compute budget since this is the fundamentally rate-limiting step in obtaining explanations, but practitioners will do well to compare methods with respect to wall-clock time as well.