# OpenReview forum: "The Out-of-Distribution Problem in Explainability and Search Methods for Feature Importance Explanations"
_NeurIPS.cc/2021/Conference — NeurIPS 2021 Poster_

### Official Review · Reviewer_abDP · 2021-06-28

**Rating:** 6
**Confidence:** 4

**Summary:**

This paper addresses a core limitation of existing feature importance methods, where importance scores are generated by comparing original examples to unrealistic counterfactuals. To solve this issue, the authors propose a solution where counterfactuals used to compute feature importance are generated by manipulating the attention mask and not by editing the actual text of the original examples. They also propose training models on the generated counterfactuals and show that it improves model robustness. Finally, they introduce a novel method producing model explanations and show that it outperforms strong baselines.


**Limitations And Societal Impact:**

This paper can potentially have a substantial positive societal impact, as understanding what features drive model decisions can uncover hidden biases and fairness issues which govern the model’s behavior.


**Main Review:**

The paper contains some interesting findings and proposes a novel search-based method which improves on existing feature-importance solutions. It is also clearly written and easy to follow through.

However, in its current state, the contributions presented in the paper are not substantial enough in my opinion. First, the realization that unrealistic counterfactuals harm our ability to estimate the effect of variables is a pillar of causal research, and there are existing methods in ML and in NLP specifically which deal with this problem in the context of causal explanations (for example, see CausaLM, INLP, CaCE and their follow-up work).

Second, the use of counterfactuals for training can be misleading. While it can improve model performance (as demonstrated in the counterfactually augmented data literature), measuring model performance on in-domain data and calling it more robust is problematic. Alternatively, counterfactually trained models should be tested on types of counterfactuals which were not seen in training. Also, counterfactual training can harm our ability to identify model weaknesses, which is a main motivation for generating model explanations in the first place.

The two main arguments raised in this paper, that the OOD-ness of counterfactuals leads to poor explanations and that training on OOD data improves robustness, are inherently causal arguments which I think already exist in the literature. While the results in the paper are surely interesting, I would like to see the authors address the existing literature on the subject and try to explain the similarities and differences in the theoretical approach. While I am pointing to one prism of thought that might be useful for analyzing the paper’s findings, there might be others. To improve the paper, I would like to see the authors address similar arguments made in the literature.


**Time Spent Reviewing:**

6

---

> ### Author Response · Authors · 2021-08-10
> **Reply to Reviewer abDP**
>
> > ...the realization that unrealistic counterfactuals harm our ability to estimate the effect of variables is a pillar of causal research, and there are existing methods in ML and in NLP specifically which deal with this problem in the context of causal explanations (for example, see CausaLM, INLP, CaCE and their follow-up work).
>
> We would be happy to discuss CausaLM, INLP, and CaCE at length, but we do not see how the existence of these works lessens our contributions. None of them discuss the social alignment of explanations and the impact of the OOD problem on their social alignment, so we believe our contributions are still novel on this front. To our understanding, these methods are not meant for use with individual data points either, which is what we are interested in explaining. Is there a particular way that you imagine the findings of these papers should inform the identification and evaluation of FI explanations for individual data points?
>
> Below we briefly outline our understanding of each paper individually.
>
> The contribution of CausaLM is to estimate the average treatment effect of features on the model output specifically when those features are hard to remove from an input without unintentionally affecting other aspects of the input (or specifically, without violating the data generating process). This could include the topic of a document for instance. When features are like "presence of adjectives", these can be removed from a document without violating the data generating process.
>
> CACE addresses the same problem as CausaLM but in a vision setting and using alternative methods.
>
> INLP removes information from representations by means of linear projections. The only way we see that this is relevant for us is if we used INLP to remove token information from a single input sequence for purposes of obtaining counterfactual inputs. But if we used this method to remove a token from a sequence, the result must be setting the embedding for that token to 0. This is equivalent to the zero-embedding condition that we test in our experiments.
>
> > ...the contributions presented in the paper are not substantial enough in my opinion.
>
> Besides the points made in the prior comment, we would suggest that a new explanation method that outperformed prior SOTA methods on widely accepted metrics with a diverse set of datasets might already be considered enough for publication. Here, we have additionally made the conceptual contribution of providing a new argument regarding the OOD problem and a new solution that still allows for test-time evaluation of individual explanations using popular ablation-based metrics.
>
> > While it can improve model performance (as demonstrated in the counterfactually augmented data literature), measuring model performance on in-domain data and calling it more robust is problematic.
>
> In lns 247-255, we conclude that the Counterfactual Trained models are more robust to counterfactual inputs than the Standardly Trained models, because they obtain a higher accuracy on data drawn from the counterfactual data distribution. We do not understand what the problem with this conclusion is. In particular, we know from this experiment that editing an input with a Replace function reduces performance to a smaller degree for CT than Standardly-Trained models -- we describe this result as CT models being more robust to data drawn from the Replace operation, or equivalently, that the counterfactual inputs are more in-distribution for CT models than ST models. Would you suggest changing our interpretation of this experiment?
>
> > Alternatively, counterfactually trained models should be tested on types of counterfactuals which were not seen in training.
>
> We do not perform Counterfactual-Training to improve model robustness to other kinds of out-of-distribution inputs, but to make the counterfactual inputs in-distribution points (see lns 187-188, 195-196), so this kind of evaluation is not necessary for our purposes.
>
> > Also, counterfactual training can harm our ability to identify model weaknesses, which is a main motivation for generating model explanations in the first place.
>
> This sounds relevant to consider. Could you suggest some references for this point? We do not know of any result that implies it is harder to identify model weaknesses for models that have any kind of data augmentation, or for our kind of Replace-function-based augmentation in particular.
>
> > To improve the paper, I would like to see the authors address similar arguments made in the literature.
>
> We draw a distinction with the most relevant past work in lns 152-155. In fact, the most relevant work makes only intuitive arguments for why it is problematic to use OOD counterfactuals with explanation evaluation. The works mentioned before (CausaLM, INLP, CaCE) do not address this problem, though we would be happy to discuss differences with these works using the extra page for camera-ready papers. We provide a new and stronger argument against using OOD counterfactuals by drawing on the concept of social alignment, and we note the advantages of our solution to this problem in lns 201-203, relative to other past solutions.

---

### Official Review · Reviewer_2JSj · 2021-07-05

**Rating:** 6
**Confidence:** 4

**Summary:**

Summary

Explanations methods based on feature importance (FI) assign an importance score to each input feature. This is often evaluated by (1) removing the k most important features and replacing it with a feature that is in some sense defined as uninformative (also referred to by “Replace” function in the paper), (2) measuring how much the confidence in the original prediction drops (= sufficiency metric), (3) a good explanation is determined by still having a high confidence in the original prediction.

The paper argues that this approach is problematic because the explanations, were some features are replaced with uninformative features, represent a counterfactual out-of-distribution (OOD) example that the model is not equipped to handle correctly. As a consequence, the paper argues, the “best” explanation, as determined by the above approach, does not necessarily provide the information the user expects, making this explanation “socially misaligned”.

The paper studies the problem and explores alternative options, which are tested on six text classification tasks.

Contributions

-	Arguments on why the current sufficiency-based evaluation of FI explanation methods is problematic
-	Investigation of which “uninformative” features are best for avoid the outlined problem (see second paragraph above)
-	Novel search-based methods for finding FI explanations


**Limitations And Societal Impact:**

-	The paper provides a good overview of the potential risks of FI explanation methods.

**Main Review:**

Strengths

-	Interesting discussion on how OOD explanations do not reflect what a user expects and how this is an issue. Considerations of how explanations actually affect users is important in order for the research field of explainable AI to head into a direction where explanations truly reflect what is needed in applications by users.
-	Systematic comparison of various FI explanation methods and different choices for selecting uninformative features. This gives interesting insights and is particularly value for anyone who wants to apply FI explanation methods in practice.
-	A new search methods that finds the best explanation iteratively (with regards to sufficiency metric), rather than taking the top k highest as features
-	Experimental results are thorough and follow good practice (e.g. 10 random seeds, error bars)

Weaknesses

-	Some questions remain open but can easily be addressed.
-	It is not clear that the insights drawn will really generate explanations that are truly more socially aligned. Of course, answering this question is in itself tricky and the paper acknowledges this in the broader impact statement

After Authors' Response
Thank you for answering my questions! I like the idea of this paper, although I do agree with some of the concerns raised by other reviewers, particularly that the story line is in parts not clear to follow. Thus, I will keep my score unchanged.

Questions

-	Line 152: Why do you say a MASK token is OOD, i.e. not seen in training? At least in pre-training of a masked LM, the MASK token is part of training.
-	Paragraph starting at line 163: Yes, model prior and random seed affect an explanation, but they should, correct? A different seed might really lead to a different model and therefore a different explanation? [see also footnote 1 below] My guess is that the main difference is: models with different random seeds also handle the same OOD counterfactual differently and how different models react to OOD counterfactuals remains intransparent, leading to the social misalignment? Do you agree? Could the argument be made clearer?

[footnote 1] This also relates to what Jacovi&Goldbert term faithfulness, in that context for an explanation to be faithful, a model with a different random seed and a different prediction also might have a different explanation and it needs to have a different explanation because it is indeed a different model, this is needed for the explanations to be faithful.

-	Line 224, “Marginalize”: I’m not sure I understood this paragraph correctly, is the following correct: For the input feature to be removed, you use BERT to replace the token at the to-be-removed feature position? E.g. in “A cat is sat on a mat” and “cat” is to be replaced, you ask BERT what other word to put there and it e.g. give you “dog”?
-	Line 267, Vanilla Gradients: Which replace function do you use there?
-	LIME vs Integrated Gradients: Why does one use the replace function attention mask and the other mask token?
-	Table 1: The difference between e.g. LIME and random search is that LIME uses the top-k features whereas random sample k at random? Sorry if I missed it, but what is the value for k? How did you choose it?

Typos, representation, minor questions etc.

-	115: explanation -> explanations
-	It is strange that the sufficiency metric is introduced twice
-	It took me a few reads of the beginning of section 5 to understand what exactly the setup is and why, maybe this could be rephrased some? For example, “where the set of input features to be removed is fixed” -> explicitly state here why this is important and needed
-	Line 290: delete “produce”
-	Line 294: delete “with”
-	Line 295: what is the regularizer here?
-	Line 334: It would be great if you could list the budget (forward/backward passes) required by the different methods that you test.
-	Table 1: Bold in last section, second last column is missing; same for last column of “Evidence Inference”



**Time Spent Reviewing:**

3

---

> ### Author Response · Authors · 2021-08-10
> **Reply to Reviewer 2JSj**
>
> > Why do you say a MASK token is OOD, i.e. not seen in training? At least in pre-training of a masked LM, the MASK token is part of training.
>
> It's true that the MASK token is seen in training. We take "OOD" to mean that data appears which is not drawn from the training distribution (ln150), and we consider the training distribution to be the data distribution for the finetuning task rather than the pre-training task. We take this view because the estimated model p(y|x) has not been trained to handle the mask tokens for purposes of classification, even though the pretrained model p(x) was trained to handle the mask tokens for purposes of token prediction.
>
> > ...for an explanation to be faithful, a model with a different random seed and a different prediction also might have a different explanation and it needs to have a different explanation because it is indeed a different model
>
> We strongly agree with this observation, and this is why we take care to write "particular trained model" in ln163, and why we evaluate methods over 10 different model seeds in each experiment. We can be more explicit about this point in Sec. 2 in the final version of the paper.
>
> > ...model prior and random seed affect an explanation, but they should, correct? A different seed might really lead to a different model and therefore a different explanation?....My guess is that the main difference is: models with different random seeds also handle the same OOD counterfactual differently and how different models react to OOD counterfactuals remains intransparent, leading to the social misalignment? Do you agree? Could the argument be made clearer?
>
> It's true that the model prior and random seed will affect downstream explanations. The question of social alignment is: do people think that FI explanations should reflect the influence of the prior and random seed? In Sec. 4, we suggest that people expect explanations to be influenced by what the model has learned, but not by other things that the model did not learn like the prior and random seed.
>
> It is an interesting observation that two models with differing random seeds may have differing explanations for the same data point. Indeed, it is intransparent why they end up having differing explanations, because the research community does not have a fine-grained understanding of how seeds influence large learned models. If this mechanism were more transparent, researchers and practitioners might develop different expectations of what should influence the explanations, because they may become comfortable with the (known) manner in which seeds influence explanations. The picture is a little clearer with model priors: If we come to a clearer understanding of how model priors influence explanations for neural models (akin to how we understand how Gaussian Process priors would influence hypothetical explanations for GP models, see Sec. 4), then we might be more comfortable with how priors influence model explanations. But the prior is the choice of the model designer, and therefore in this hypothetical we are supposing people would be more comfortable with model designers directly influencing the values of feature importance explanations, which may not be the case. In our paper, we argue that people expect FI explanations to reflect what the model has learned from data and not the prior and seed (i.e., choices of the designer). So we suspect transparency is not the deciding factor. Rather, it is the discrepancy between what people expect to influence FI explanations and what actually influences them (represented in Fig 1, and described in lns 185-186).
>
> Thanks for the questions -- given our discussion, we should try to clarify in lns 163-186 that the intransparency is not fundamentally the issue, although people may also be dissatisfied with the intransparency.
>
> > “Marginalize”: I’m not sure I understood this paragraph correctly, is the following correct:...
>
> Thanks for checking on this. Your description is close to complete. We do ask BERT what word (or, specifically, token) should go in a position where a token has been removed, but we do this many times. Each time, we sample a different token from the distribution given by BERT over possible missing tokens. We take a weighted average over the model outputs for each of these imputed data points, weighted by the probability of drawing the token that went into the input according to BERT (this is marginalizing the model outputs over the distribution over missing tokens given by BERT).
>
> We describe the approach using greater mathematical detail in Supplement A.1.
>
> > Vanilla Gradients: Which replace function do you use there?
>
> This is actually the one explanation method that does not use any counterfactuals, and hence no Replace function is needed. The gradient of the model output is computed w.r.t. the observed model input, which in this case is the d x t matrix representing d dimensional embeddings for a sequence of length t. To get a single value per token, we sum across the d dimension. We can clarify this in the final version of the paper.
>
> > LIME vs Integrated Gradients: Why does one use the replace function attention mask and the other mask token?
>
> Good question. We use Attention Mask with LIME following the favorable results for Attention Mask in Sec. 5. The way that Int-Grad works makes it difficult to use this Replace function. Specifically, Int-Grad computes the model output at points between the counterfactual and the observed data point. Since the attention mask in a transformer can only take binary values, we can't really interpolate between a attention mask full of ones and one that is all zeros. We are forced to interpolate between the MASK token embeddings and the embeddings of the observed tokens in the input. Here, we use the MASK token here rather than the all zero embedding given our results in Sec. 5, and contrary to the original suggestion of Sundararajan et al.
>
> > Line 295: what is the regularizer here?
>
> It is the absolute difference between the expected sparsity of $e_t$ and the target sparsity. The expected sparsity of $e_t$ is the sum of elements in $s_t$. The target sparsity would be, for example, 2 if the sequence length is 20 and the sparsity level is .1 (see lns 137-139).
>
> > Line 334: It would be great if you could list the budget (forward/backward passes) required by the different methods that you test.
>
> Thanks for the suggestion! This sounds helpful and we can do so with the extra page in the camera-ready version.
>
> > Table 1: Bold in last section, second last column is missing; same for last column of “Evidence Inference”
>
> This is a good observation -- actually, per lns 342-343, we do not bold any numbers when there are no statistically significant pairwise comparisons.

---

### Official Review · Reviewer_Lhz2 · 2021-07-15

**Rating:** 3
**Confidence:** 4

**Summary:**

The paper considers different aspects of feature-importance-based explanations. In particular, the authors propose a ‘counterfactual training’ approach to deal with observations that are out of the original training distribution (OOD smaples). Moreover, the authors illustrate different approaches to replace input features with a pre-defined baseline, i.e., a common approach to assess the quality of an attribution/explanation method, which however may lead to OOD samples. Finally, the authors propose search algorithms to find binary-valued, feature-based explanations that have been shown to improve relevant test measures.

**Ethical Concerns:**

There are no apparent ethical concerns regarding this work.

**Limitations And Societal Impact:**

The authors have addressed some general limitations of local explanation methods. However, there are several more specific limitations and problems that the authors have not addressed yet (see detailed comments).

**Main Review:**

The paper discusses interesting aspects of feature-attribution-based explainations and introduces a novel approach to deal with the out-of-distribution problem of ablated observations. The use of search methods as an alternative to local (linear) approximations for the identification of important features seems interesting.

Unfortunately, there are several concerns regarding the overall quality and contribution of the paper, which I outline below:
The main contributions of this paper (line 72) do not form a consistent story line. While the first two contributions consider the evaluation of local attribution methods, the third contribution (search methods for local explainability) is somewhat isolated from the rest of the paper.
The term ‘counterfactual’, as used in this paper, does not adhere to common definitions in the literature. The authors speak of a counterfactual as an input, ‘which has its important features replaced with a user-specified (uninformative) feature’ (line 45). In general, however, a counterfactual explanation is considered the smallest change made to an input vector to influence its prediction. The authors might want to talk of ‘ablated inputs’ (or similar) instead.
In the related work section, the authors should also consider the recent paper of Haug et al. [1], when talking about the effect of different replace (baseline) methods.
The specified main objective (line 136) is to generate explanations that provide good sufficiency/comprehensiveness statistics in the evaluation. However, existing work on local attributions already specifies much stricter and more meaningful properties (e.g. Lundberg et al. [2]). The authors should argue, why their objective should be preferred.
The authors acknowledge the (negative) effect of out-of-distribution samples in the evaluation of local explanations. However, rather than improving the explanation or evaluation process, the authors manipulate the model training to obtain ‘expected behaviour’ (line 162). This is a highly questionable approach, as the very purpose of an explanation technique is to communicate what a given (!) black-box model does/has learned. In fact, the authors' approach to bring ablated inputs in line with the training distribution, is to alter the original training distribution (and therefore the trained black-box model). In this context, the proposed ‘counterfactual training’ is an approach to make predictive models more robust, but, in my opinion, it is not a meaningful technique to improve the quality of explanations.
The description of the ‘Replace’ functions (lines 221-230) is insufficient. For example, it remains unclear what a ‘MASK token’ is. Besides, the different ‘Replace’ functions are compared with respect to random explanations. The differences observed in Figure 2 might be attributed to this random sampling. The authors should therefore discuss, why random explanations are to be preferred here over real explanations (e.g. LIME).
In the experimental description, the authors describe that they ignore negative attributions when selecting the top $k$ features (line 262). However, (strongly) negative attributions can indicate important features as well (see SHAP and LIME, for example). The authors might want to reconsider this approach.
The advantage of the proposed search methods for local attribution finding remains unclear. Other than empirically improved sufficiency and comprehensiveness scores, the authors fail to provide any sensible properties or strong guarantees on the generated attributions. In fact, the search methods only yield 0/1-indications of important features and thus contain much less information than existing attribution methods (which usually generate scalar weights and thus also allow a ranking of features).
Finally, in the discussion of the experiments (line 357), the authors find that the sufficiency/comprehensiveness scores are often much worse when using the proposed counterfactual training. This contradicts the earlier recommendation to use counterfactual training ‘whenever one intends to create feature-importance-based explanations’ (line 253).

[1] Haug, Johannes, et al. "On Baselines for Local Feature Attributions." arXiv preprint arXiv:2101.00905 (2021).
[2] Lundberg, Scott M., and Su-In Lee. "A unified approach to interpreting model predictions." Proceedings of the 31st international conference on neural information processing systems. 2017.



**Time Spent Reviewing:**

4

---

> ### Author Response · Authors · 2021-08-10
> **Reply to Reviewer Lhz2**
>
> > The main contributions of this paper (line 72) do not form a consistent story line.
>
> In our view, our work on local attribution methods is dependent on our work on the OOD problem, because in order for local attribution methods to be trustworthy and faithful, the OOD problem must first be resolved. We would be uncomfortable omitting Sec. 3 from the paper and subsequent experiments with Counterfactual Trained models, knowing that the results there are important for understanding the results in Sec. 6.
>
> Additionally, as noted in ln207-208, we have to pick a Replace function to use in the attribution methods, and therefore we evaluate Replace functions (Sec. 5) before evaluating attribution methods (Sec. 6). For example, we use the Attention Mask Replace function with LIME in later experiments because we found it to outperform the default Slice-Out function that is more commonly used with LIME.
>
> > The term ‘counterfactual’, as used in this paper, does not adhere to common definitions in the literature...a counterfactual explanation is considered the smallest change made to an input vector to influence its prediction.
>
> We note that the term 'counterfactual explanation' never appears in the paper, and hence we do not think there should be any confusion that our feature-importance explanations are actually counterfactual explanations.
>
> See lns 41-43 for why we invoke the term 'counterfactual.'  We think that our usage of the term 'counterfactual input' is in accordance with the usage of the term 'counterfactual' in the broader fields of AI and causal inference, where the term denotes alternative variable assignments to the observed variable assignments (see ref [38] in our paper).
>
> > In the related work section, the authors should also consider the recent paper of Haug et al.
>
> Thanks for the reference! We missed this concurrent work as it was published just two months prior to the submission deadline, but we will be sure to add it to the related works section.
>
> > The specified main objective (line 136) is to generate explanations that provide good sufficiency/comprehensiveness statistics in the evaluation. However, existing work on local attributions already specifies much stricter and more meaningful properties...Other than empirically improved sufficiency and comprehensiveness scores, the authors fail to provide any sensible properties or strong guarantees on the generated attributions.
>
> Test-time ablation metrics like sufficiency and comprehensiveness have been widely adopted in published work on explainability (see paper references [2, 15, 42, 28]). Due to space constraints in the original paper, we did not re-justify the use of these metrics, but we can do so here.
>
> These metrics show whether explanation methods actually identify the evidence used in model decisions. It seems we may disagree about the relative value of empirical results on these metrics versus theoretical guarantees for other properties. While the theoretical properties provided for methods like SHAP and Int-Grad are nice to have, it would be a problem if these methods could not actually identify the evidence used in model decisions. In fact, this is what we observe. It seems like Integrated Gradients heavily underperforms on these metrics (but not LIME, which also does not have theoretical guarantees). We would like to suggest that Int-Grad is simply not a good explanation method with our models and data, per these widely accepted metrics. This is not especially surprising to us since Int-Grad was developed primarily to fulfill some theoretical desiderata, and there are no quantitative evaluations of explanation quality in the original paper. While it might be nice that the importance attributions of Int-Grad sum to the difference between model outputs for the observed and baseline data points -- an example of one theoretical property -- this is not very helpful when we are actually trying to find out what features counted as evidence for a particular model decision, as shown by our experiments. This finding is consistent with other work that is critical of gradient-based explanations including Int-Grad (e.g. "Sanity Checks for Saliency Maps", Adebayo et al. 2018; “A Benchmark for Interpretability Methods in Deep Neural Networks”, Hooker et al. 2019;
>  “Are Visual Explanations Useful? A Case Study in Model-in-the-Loop Prediction”, Chu et al. 2020).
>
> > ...the authors manipulate the model training to obtain ‘expected behaviour’ (line 162). This is a highly questionable approach, as the very purpose of an explanation technique is to communicate what a given (!) black-box model does/has learned
>
> We seem to be miscommunicating here, because we agree with your stated purpose of explanation techniques! This is precisely what we wrote in lns 163-167. We mean to suggest that, for standardly trained models, the OOD problem implies that there can be no socially aligned feature importance explanations. The only way to get socially aligned feature importance explanations is to manipulate the training process for the model being explained, by aligning the train and test-time distributions (lns 187-188). Therefore, if we want to keep using test-time ablation metrics, we have to adjust the model training distribution. Of course this will necessarily result in a different model that is being explained, meaning this is a choice that must be made in advance of the decision to explain a model prediction.
>
> > The description of the ‘Replace’ functions (lines 221-230) is insufficient. For example, it remains unclear what a ‘MASK token’ is.
>
> We are sorry we had to heavily compress this method due to space constraints. An expanded version of the descriptions is included in Supplement A.1.
>
> The MASK token is commonly known in NLP to be the token used to mask out tokens during pretraining of transformers such as BERT and RoBERTa. We thought that by noting we use these models prior to this description, it would be clear what MASK token referred to.
>
> > ...the different ‘Replace’ functions are compared with respect to random explanations. The differences observed in Figure 2 might be attributed to this random sampling. The authors should therefore discuss, why random explanations are to be preferred here over real explanations (e.g. LIME).
>
> As described in lns 206-208, we have to pick a Replace function to use in the attribution methods, so we should first evaluate which Replace function would be best to use. If we used an explanation method to get explanations, we would have to pick a Replace function before evaluating all of the Replace functions, and this could bias the results for or against a certain Replace function. We see the only way to make a fair comparison is to average results over a sizable number of random explanations.
>
> > In the experimental description, the authors describe that they ignore negative attributions when selecting the top k features (line 262). However, (strongly) negative attributions can indicate important features as well (see SHAP and LIME, for example).
>
> It’s true that strongly negative attributions are important, but they are supposed to be evidence against the predicted label that is being explained. In Sufficiency and Comprehensiveness, we are interested in keeping or removing the evidence for the predicted label, not against it. Therefore, we never keep or remove features with negative attributions. This choice leverages the fact that salience methods can return signed attribution scores to improve the scores these methods obtain.
>
> > ...the search methods only yield 0/1-indications of important features and thus contain much less information than existing attribution methods (which usually generate scalar weights and thus also allow a ranking of features).
>
> Regarding the rank-ordering point, we agree that rank-ordering has some useful properties, like being able to quickly identify what the top-k features are for various k. In Supplement C, we suppose that a user will be interested in multiple feature sets, according to various k values. We show that when the user wants four such sets, our search method still outperforms LIME in 15 of 24 comparisons, while controlling for the total compute budget both methods are allowed to use. Hence, our method is still preferable for users who are interested in multiple sizes of explanation (within the same time limit).
>
> However, we emphasize that if a user asks “What are the top-k features counted as evidence for the model's prediction?”, then Parallel Local Search provides the best answer, more so than any of the salience-based methods we considered. Are there other reasons you prefer scalar weights besides what is accounted for here?
>
> > ...the authors find that the sufficiency/comprehensiveness scores are often much worse when using the proposed counterfactual training. This contradicts the earlier recommendation to use counterfactual training ‘whenever one intends to create feature-importance-based explanations’ (line 253).
>
> Since we do not recommend to use whatever training algorithm leads to the best explanation scores, this finding does not contradict our earlier recommendation. As described in lns 195-197, we recommend that researchers use Counterfactual-Trained models when creating FI explanations because this approach protects against the OOD problem, irrespective of the Suff/Comp scores that will follow from using CT models. Due to space constraints we did not include an additional section emphasizing this point: If researchers want socially aligned FI explanations, they should use CT models. Once a trained model has been obtained, one should use the explanation method that produces the best Suff and Comp scores. We will be able to emphasize this point using the extra page for camera-ready papers.

---

### Official Review · Reviewer_NMtf · 2021-07-16

**Rating:** 5
**Confidence:** 3

**Summary:**

This paper investigates feature important methods in NLP. The authors argue that counterfactual inputs should be in-distribution, and propose training on counterfactual inputs to reduce this distribution shift. They also investigate the best approach to producing counterfactuals and the best search method for finding FI explanations, finding that one of their proposed methods does best.

**Main Review:**

This paper addresses an important problem and has the potential to be a strong paper. It is somewhat original and includes many experiments. However, overall it feels unpolished and does not yet seem ready for publication.

The paper claims to make a new argument for why OOD counterfactuals are bad; namely, that they are socially misaligned. I strongly agree with the conclusion that OOD counterfactuals can be misleading for interpretability, as other authors have pointed out before. However, I don't feel like the argument that they're socially misaligned really adds anything.

The authors then propose training on counterfactual inputs to reduce distribution shift. This basic idea seems natural and reasonable to me. However, there are a few cases where they make claims about what approach to producing counterfactual inputs is "best" by using accuracy as a proxy for how in-distribution examples are. I think this is kind of misleading; accuracy is affected both by how OOD the data is and by how much actual information is lost (e.g. you could train on counterfactuals where information is removed, and this would then be not OOD but still have lower accuracy).

Moreover, I think it's unclear how I should think about explanations from counterfactually-trained models. It seems plausible but not obvious to me that I should now basically trust FI explanations from counterfactually-trained models (whereas FI explanations from non-CT models are misleading because of the OOD gap). To the extent that you're proposing a new approach to explaining models, you should explain to me in what sense it's an explanation, why I should trust it, etc.

Relatedly, the authors use metrics of sufficiency and comprehensiveness, which seem to capture meaningful aspects of how good an explanation is. However, I'm not convinced they capture everything we care about. There are also some mysterious results, such as how some "strong" baseline methods do terribly according to these metrics. This makes me suspicious that either these baselines are not strong, or that these metrics are unreliable.

In general, I think some of the results are potentially interesting, but in many cases I'm not sure how much to trust them or how I should best interpret them. I think a clearer exposition would go a long way toward these problems, which I think make this work not yet publishable.

---------------
Update: I've read the rebuttal. While it clarifies some minor points, overall I still don't think this work is ready for publication, so I'll choose to keep my score.

**Time Spent Reviewing:**

3

---

> ### Author Response · Authors · 2021-08-10
> **Reply to Reviewer NMtf**
>
> > I don't feel like the argument that they're socially misaligned really adds anything
>
> Relative to past work, we give the strongest argument against using OOD counterfactuals in creating/evaluations FI explanations. Historically people have made arguments based on intuition or basic machine learning principles like avoiding distribution shift (described in lns 152-155). Evidently these arguments have not persuaded the larger community of this problem, because FI explanations are typically constructed and evaluated using OOD counterfactuals (references [46, 59, 32, 2, 15, 42, 28] in paper). We hope that this new, strong argument based on social alignment will help lead to a decline in its prevalence.
>
> > ...there are a few cases where they make claims about what approach to producing counterfactual inputs is "best" by using accuracy as a proxy for how in-distribution examples are. I think this is kind of misleading; accuracy is affected both by how OOD the data is and by how much actual information is lost (e.g. you could train on counterfactuals where information is removed, and this would then be not OOD but still have lower accuracy).
>
> In these experiments, we remove the same information from inputs in every condition (described in ln210-214). Since we control for what information is removed across the conditions, the only reason that the conditions in our Sec. 5 experiment would differ is the OOD-ness of the counterfactuals, which is consistent with your suggestion that accuracy declines due to either OOD-ness or information removal.
>
> We also train models on the counterfactual data, matching your suggestion, with results shown in Fig 2 RHS and described in lns 247-255. Here, the difference in conditions is largely erased, though we still observe a statistically significant difference between Attention Mask and Zero Embedding (favoring Attention Mask).
>
> > It seems plausible but not obvious to me that I should now basically trust FI explanations from counterfactually-trained models (whereas FI explanations from non-CT models are misleading because of the OOD gap). To the extent that you're proposing a new approach to explaining models, you should explain to me in what sense it's an explanation, why I should trust it, etc.
>
> Given space constraints we did not attempt to justify feature-importance explanations themselves, but our explanations are explanations in the same sense that normal FI explanations are. More specifically, good performance on Sufficiency and Comprehensiveness metrics shows that our explanations identify the components of an input that can account for the original model prediction (Suff), as well as all of the components of the input that did contribute to the prediction (Comp). Our explanations are in fact more trustworthy than standard FI explanations since the OOD problem is mitigated in our approach.
>
>
> > ...authors use metrics of sufficiency and comprehensiveness, which seem to capture meaningful aspects of how good an explanation is. However, I'm not convinced they capture everything we care about.
>
> Thanks for raising this concern. Is there any particular shortcoming you think we should discuss? These are widely used metrics in past published work (see paper references [2, 15, 42, 28]).
>
> > There are also some mysterious results, such as how some "strong" baseline methods do terribly according to these metrics. This makes me suspicious that either these baselines are not strong, or that these metrics are unreliable.
>
> We would like to point out that LIME often does well, in many cases almost as well as our parallel local search, so this is a strong baseline (see Table 1, especially MultiRC results). It seems like Integrated Gradients is the notable underperformer. We would like to suggest that Int-Grad is simply not a good explanation method with our models and data, per these metrics. This is not especially surprising to us since Int-Grad was developed primarily to fulfill some theoretical desiderata, and there are no quantitative evaluations of explanation quality in the original paper. This finding is consistent with other work that is critical of gradient-based explanations including Int-Grad (e.g. "Sanity Checks for Saliency Maps", Adebayo et al. 2018; “A Benchmark for Interpretability Methods in Deep Neural Networks”, Hooker et al. 2019;
>  “Are Visual Explanations Useful? A Case Study in Model-in-the-Loop Prediction”, Chu et al. 2020).

---

### Author Response · Authors · 2021-08-10
**General Response**

We thank the reviewers for their comments and time investment in reviewing our work. We appreciate that reviewers found that the paper “addresses an important problem” (R1),  provides a “novel approach” (R2), and contributes “interesting insights” (R3) and “interesting findings” (R4). We were particularly glad to read that our “experimental results are thorough and follow good practice” (R4).

We have some questions and comments that we will give for each reviewer’s comments separately below. Looking forward to discussing the merits and limitations of the work!

---

### Decision · Program_Chairs · 2021-09-27

**Decision:**

Accept (Poster)

**Comment:**

The reviews for this paper were mostly borderline, with one review strong negative. While the reviewers thought that the paper addresses an important problem in a novel way, and provides interesting insights, the author response did not lead them to change their scores.

Despite this lukewarm assessment, I believe the paper makes contributions that likely merit acceptance: (1) Tackling an important problem in generating and evaluating explanations in NLP and ML more broadly (the OODness of feature importance attribution methods that are based on masking, ablation, etc., since they produce counterfactual/OOD examples); (2) providing a clear approach to mitigate this problem (adding counterfactuals at training time), which can work well (little drop compared to standard models); (3) evaluating various methods to create explanations and finding that some work well in the new counterfactually trained models (achieve good sufficiency/comprehensiveness scores, which are standard).

Given these contributions, and considering that no paper is perfect, I recommend acceptance. Since my recommendation goes somewhat against the general sense of the reviews, I provide here a detailed discussion of the main issues raised by the reviewers, how the authors responses, and my opinion of the issue. In any case, I urge the authors to take these issues into account in their next revision.

**Comments by Reviewer NMtf**
- argument on social alignment doesn't add much. Authors: it's another perspective on why FI OODness is problematic, and may help. AC: natural application of the social alignment argument from Jacovi and Goldberg to another class of explanations.
- Using accuracy to assess distribution shift is misleading, can conflate OODness and information removal. Authors: remove the same information in all conditions, difference is due to OODness. AC: agree that accuracy change between iid and ood data is a reasonable metric of OODness.
- Not convinced about sufficiency and comprehensiveness metrics. Authors: These are standard in the literature. AC: agree, these are standard faithfulness metrics. What's missing here is an evaluation of plausibility, for example by comparing with human-annotated explanations (highlights, rationales).
- Baseliens do terrible. Authors: LIME does well, IG is likely a poor method. AC: It's tricky to compare when the benchmark is not common. It could have been better to use standard datasets from explanations in NLP, where prior results are available. But, LIME results are indeed quite good.

**Reviewer Lhz2**
- The contributions do not form a consistent story line. Authors disagree and explain that the part on local attributions depends on fixing the OOD problem of explanations. AC: agree, the first part is a nice contribution in itself and it makes the rest possible.
- Counterfactual is an incorrect term here, since a counterfactual explanation means something different. Authors:  did not use counterfactual explanation. AC: best to clarify that counterfactual input is not to be confused with the technical term of counterfactual explanation.
- questions the sufficiency/comprehensiveness metrics and prefers more meaningful metrics like SHAP. Here I agree with the authors that it is useful to apply these empirical metrics, which are standard in the literature, even if they do not have some nice theoretical properties like SHAP.
- Changing the training distribution is questionable. Authors: to use FI explanations, have to align training and test distributions, so if we want to keep test time metrics, must change training distribution. AC: indeed, I was also concerned by changing the training distribution and hence the model, but was pleased with the minor drop in accuracy, at least for some of the methods. However, it's possible that a model trained on the original distribution arrives at decisions in a different way than a model trained on the counterfactually-modified distribution, but they have similar performance. This is a hard problem to get around, but one that should be discussed.
- Search methods provide binary importance while existing methods provide scalar scores. Authors argue that their method still brings useful rankings (15 out of 24 vs LINE) and that for selecting features it works best. AC: not convinced by the author response, as 15/24 seems weak, and also as scalar weights can indeed be useful, both for interpretation (users might prefer them) and for downstream tasks (training with explanations, etc.).
- Sufficiency and comprehensiveness are often worse when using the proposed counterfactual training. Authors explain they recommend using CT regardless of these metrics, because using CT guards again OODness. AC: while I agree with the idea in principle, I'm unconvinced by the argument. If using a standard model leads to better explanations according to the metrics in question, then the argument for using CT grows much weaker.

**Reviewer 2JSj**
- Had some questions that seemed to have been answered.
- Not clear that the explanations will truly be more socially aligned. AC: did not see a clear answer to this. The answer to the issue of social alignment and how different models react to OOD counterfactuals is part of the story, but probably not all.

**Reviewer abDP**
- there are existing method to deal with the problem of estimating effects in the case of counterfactuals (mentioning specific studies: CausaLM, INLP, CaCE). Authors: don't see how these particular studies lessen the contribution, since they don't discuss how OODness affects social alignment. Also say that the mentioned methods are not meant to use with individual data points. AC: agree. Those should probably be discussed, but are not at the core of the current work.
- Counterfactual training can be misleading in terms of evaluation. Authors argue that in their case it makes sense, because the CT models work better on OOD data, and they're not interested in different kinds of OODness. AC: agree, this actually seems reasonable for the present use case of evaluating explanations, if not for robustness more broadly. The authors should clarify their goals in this evaluation and distinguish from general model robustness.